**EMBO** *reports*

# Symbionts with eroded genomes adjust gene expression according to host life-stage and environment

Ana S P Carvalho[1,2], Sinah T Wingert [ID][1,2], Roy Kirsch[1], Heiko Vogel[1], Gregor Kölsch[3,4] & Martin Kaltenpoth [ID][1,2][✉]

## Abstract

**Symbiotic bacteria in long-term host associations frequently undergo extreme genome reduction. While they retain genes beneficial to the host, their repertoire of transcription factors is severely reduced. Here, we assessed whether genome-eroded symbionts can still regulate gene expression by characterizing the transcriptional responses of obligate symbionts in reed beetles to different temperatures and host life stages. These symbionts feature a small genome (~0.5 Mb), encoding for 9–10 essential amino acid biosynthesis pathways, 0–2 pectinases, and 4–5 transcription factors. We found that the symbionts respond to winter conditions by upregulating a heat-shock sigma factor and downregulating translation machinery. Across life stages, symbionts adjusted gene expression to meet the hosts' nutritional demands, upregulating amino acid biosynthesis in larvae, while expression and activity of host and symbiont enzymes involved in plant cell wall breakdown increased in the folivorous adults. In addition, the regulation of symbiont cell morphology genes corresponded to cell shape differences across life stages. Thus, reed beetle symbionts may use their few transcription factors to respond to the host's environment, highlighting the regulatory potential of long-term coevolved symbionts despite severely reduced genomes.**

**Keywords** Cell Shape; Genome Erosion; Pectinase; Symbiosis; Transcriptional Regulation
**Subject Categories** Evolution & Ecology; Methods & Resources; Microbiology, Virology & Host Pathogen Interaction

## Introduction

Animals evolved in a microbial world, and their ecology and diversification are often shaped by their microbial partners (Margulis and Fester, 1991; McFall-Ngai et al, 2013). These interactions with bacteria can cover a wide range of relationships, ranging from pathogenicity to mutualism (Drew et al, 2021). The latter can be crucial for animals to expand their ecological niche space. In particular, many insects can only exploit a strictly herbivorous diet in symbiosis with a microbial partner (Cornwallis et al, 2023; Hansen and Moran, 2014; Mason et al, 2019). This is due to the insects' inability to synthesize essential amino acids and vitamins, which are in low abundance in many plant tissues, as well as the need for specialized sets of enzymes to degrade the plant cell wall (PCW). Thus, symbionts of herbivorous insects often supplement essential amino acids, vitamins, and/or provide digestive enzymes (Cornwallis et al, 2023; Mason et al, 2019; Salem and Kaltenpoth, 2022).

Many symbioses between bacteria and herbivorous insects are ancient and reflect a long history of co-evolution in which the bacteria have become host-restricted (Cornwallis et al, 2023). In such systems, bacteria commonly suffer drastic genome reduction (McCutcheon and Moran, 2012). As a result, they retain a minimal set of genes required for symbiont maintenance and benefit provisioning (Chong et al, 2019; Fisher et al, 2017; Manzano-Marín and Latorre, 2016; McCutcheon and Moran, 2012; Reis et al, 2020; Williams and Wernegreen, 2015), while mechanisms for regulation of gene expression of specific pathways are often lost (Chong et al, 2019; Manzano-Marín and Latorre, 2016; Wernegreen, 2017). Consequently, the regulation of symbiont-provided benefits is often attributed to the host via the control of symbiont titers (Whittle et al, 2021). However, the reduced genomes of many symbiotic bacteria retain a small number of transcription factors (TFs). Indeed, in the symbiont of the aphid *Schizaphis graminum*, a functional copy of the TF MetR upregulates the transcription of *metE*, responsible for methionine synthesis, in response to elevated homocysteine levels (Moran et al, 2005). However, the extent of gene regulatory capabilities of genome-eroded symbionts remains poorly explored, often due to the experimental and genetic intractability of these symbioses.

Here, we take advantage of a 75–100-Mya-old symbiosis (Kölsch and Pedersen, 2008, 2010; Reis et al, 2020) with ecological

[1]Department of Insect Symbiosis, Max Planck Institute for Chemical Ecology, Hans-Knöll-Str. 8, 07745 Jena, Germany. [2]Evolutionary Ecology, Institute for Organismic and Molecular Evolution (iomE), Johannes Gutenberg University, Hanns-Dieter-Hüsch-Weg 15, 55128 Mainz, Germany. [3]Molekulare Evolutionsbiologie, Institut für Zoologie, Universität Hamburg, Martin-Luther-King-Platz 3, 20146 Hamburg, Germany. [4]Lernwerft Club of Rome Schule Kiel, Skagerrakufer 5, 24159 Kiel, Germany.
[✉]E-mail: kaltenpoth@ice.mpg.de

characterization of the host and presumed life-stage-specific benefits conferred by the symbionts (Kleinschmidt and Kölsch, 2011; Reis et al, 2020), to test whether such benefits can be regulated at the level of symbiont transcription. Reed beetles (Coleoptera: Chrysomelidae; Donaciinae) acquired their symbiont (Gammaproteobacteria; Enterobacterales; Enterobacteriaceae; described as 'Candidatus Macropleicola muticae' for the host species Macroplea mutica) (Kölsch et al, 2009) in the Cretaceous. The bacteria subsequently experienced genome erosion, resulting in a small genome of around 0.5 Mbp. These beetles exhibit a semi-aquatic lifestyle, with adults and larvae consuming different diets. Donaciinae adults are active during spring and summer, when they feed on the leaves or flower parts of their aquatic host plant. The larvae, on the other hand, can be found throughout all seasons feeding on the root sap of the same plant under water, as the development of these beetles lasts one to three years (Bieňkowski and Orlova-Bieňkowskaja, 2004; Buchner, 1965; Reis et al, 2020; Stammer, 1935). Thus, these beetles face a dual challenge: growing on an amino acid- and vitamin-poor diet as larvae and the need to break down the PCW of leaf tissues as adults. Correspondingly, the symbionts of the reed beetles encode the biosynthetic pathways of 9–10 essential amino acids as well as the B-vitamin riboflavin (Reis et al, 2020), which can support larval growth (Douglas, 2006). In addition, in species that feed on pectin-rich host plants, the symbionts encode for one or two plant cell wall degrading enzymes (PCWDEs), i.e., pectinases of the glycoside hydrolase family 28 (GH28), that enable the breakdown of pectin in the PCW (Reis et al, 2020). However, in species that secondarily switched to feeding on pectin-poor host plants in the order Poales, the symbionts lost the pectinases from their genomes (Reis et al, 2020). The different benefits provided by the symbionts conform with different symbiont localizations across life stages: the symbiont is found intracellularly in foregut-associated caeca of larvae, whereas in adult beetles it is localized in the lumen and within epithelial cells of modified Malpighian tubules (Stammer, 1935).

Here, we used transcriptional profiling, enzymatic assays of digestive capabilities, and fluorescence microscopy of symbiont cell shape to assess whether the symbiont of the Donaciinae responds to environmental factors (different temperatures) in the larvae of Donacia marginata, and to the different requirements of host larvae and adults across four different Donaciinae species—Donacia marginata, Donacia thalassina, Donacia versicolorea and Macroplea mutica. For simplicity, the symbionts will be named symbDMAR, symbDTHA, symbDVER and symbMMUT throughout the manuscript, indicating the respective host species by a four-letter code. We find that the symbionts exhibit specific transcriptional responses to changes in temperature and host life stages. In the latter case, gene expression patterns correspond to the presumed benefits of amino acid supplementation during larval stages, and pectinase provisioning during host adulthood, which is further supported by the activity and upregulation of beetle-encoded PCWDE in adult guts. Furthermore, we observed symbiont cell shape differences across host life stages, coinciding with differential expression of individual genes involved in bacterial cell wall biosynthesis and regulation of cell division. These differences may represent adaptations to the symbiont's life-stage-specific roles in nutritional supplementation to the host and/or the different environments the symbiont is subjected to in larval and adult

symbiotic organs. These results provide rare insights into the regulatory potential of symbionts with a minimal gene set, highlighting the possibility to retain some metabolic flexibility despite a long co-evolutionary history and a host-restricted lifestyle.

# Results and discussion

## Symbionts respond to temperature differences

The Donaciinae develop as larvae through multiple seasons and as such experience a wide range of temperatures throughout their development (Bieňkowski and Orlova-Bieňkowskaja, 2004). Given the severely reduced set of transcriptional regulators in the symbiont genome, we set out to test whether the symbionts can still respond to different temperatures with changes in gene expression. In order to test this, we used D. marginata, as this is the only species for which the in-house rearing of the beetle and its host plant was established. Larvae of D. marginata reared in semi-field conditions (see "Methods") were transferred to an incubator and kept either at 12/8 °C or at 22/14 °C day/night temperature (hereafter cold and warm conditions) for a month, after which the symbiotic organs were dissected for RNA extraction and sequencing. After differential gene expression analysis of the symbiont, we obtained 25 and 24 genes that were differentially upregulated in cold and warm conditions, respectively (Dataset EV1). There was no significant enrichment for any particular KEGG pathway for genes differentially expressed in either condition (Dataset EV2). However, most of the genes differentially expressed across these conditions were involved in genetic information processing, mainly in translation (Fig. 1). The majority of genes differentially expressed in this category were upregulated by the symbionts under warm conditions. Conversely, genes involved in transcription, e.g., the transcription elongation factor greA and the transcription antitermination protein nusB, and the heat shock alternative sigma factor rpoH were upregulated by the symbionts under cold conditions.

The sigma factor RpoH regulon can be induced not only upon heat shock, but also upon other stresses such as carbon starvation (Jenkins et al, 1991), osmotic shock (Bianchi and Baneyx, 1999) and pH changes (Taglicht et al, 1987). Here, our results suggest that rpoH expression is induced by prolonged exposure to low temperatures. In E. coli, the post-translational regulation of RpoH is mediated by degradation via a membrane-bound metalloprotease, FtsH (Tomoyasu et al, 1995). Although the symbionts lack the ftsH gene, they encode for rseP, another membrane-bound metalloprotease that is known to regulate a stress-response sigma factor (Konovalova et al, 2018). The protease rseP is downregulated by the symbiont in cold conditions, which might allow for the upregulation of rpoH and its regulon. In addition, symbionts of larvae kept in cold conditions show the upregulation of chaperones such as groEL and the periplasmic chaperone hlpA, as well as typA, a protein putatively required for growth at low temperatures (Pfennig and Flower, 2001). The upregulation of chaperones by endosymbionts with small genomes upon experiencing stressful conditions is also seen in Stammera, which upregulates chaperone expression during its extracellular stage required for symbiont transmission (García-Lozano et al, 2024). Altogether, several genes known to be regulated by RpoH, such as groEL, nusB, ribH and

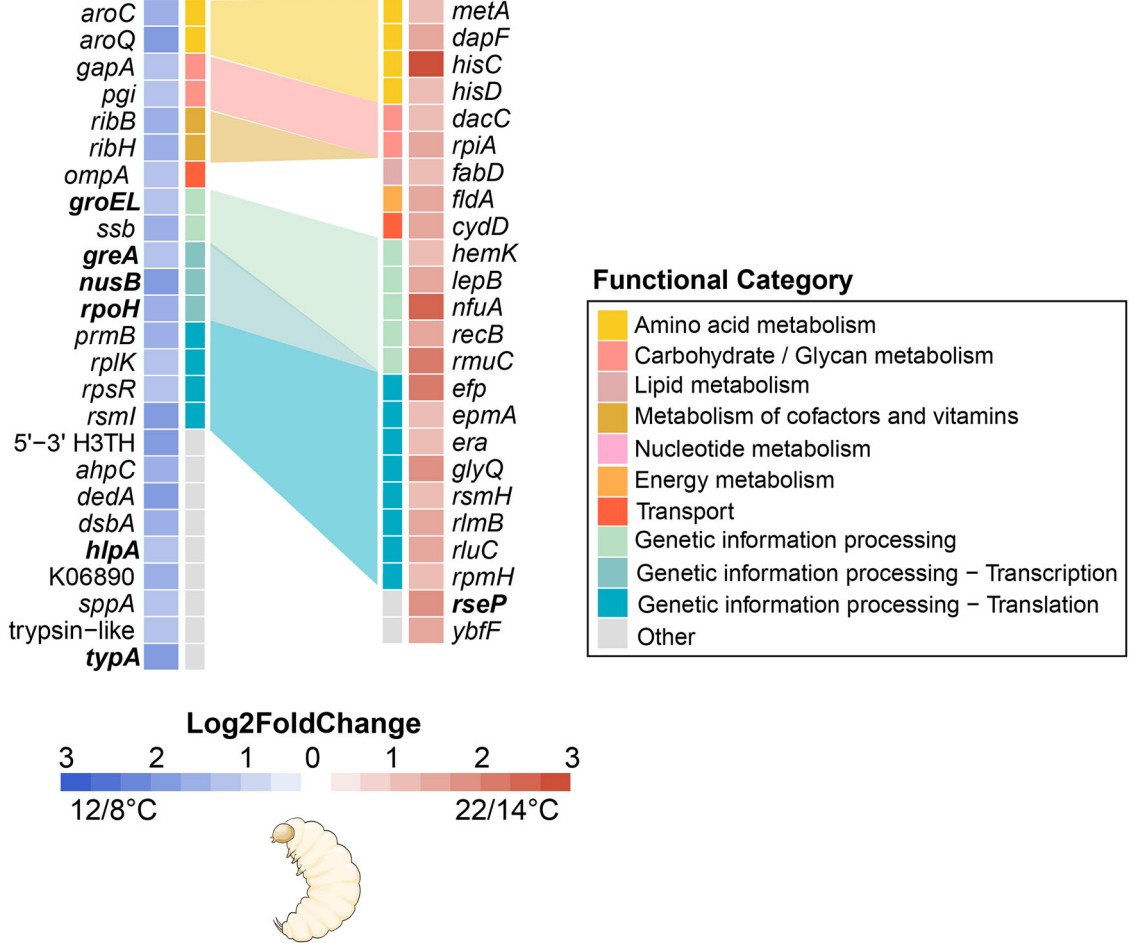

**Figure 1. Symbiont transcriptional response to different temperatures in reed beetle larvae.**

Base 2 logarithm of the fold change of genes differentially expressed by the symbionts of *D. marginata* larvae kept in warm and cold conditions, respectively (Wald test with Benjamini−Hochberg correction for multiple testing, $n = 2$ for cold condition, $n = 3$ for warm condition). Functional categories were manually curated after KEGG and InterProScan annotation. Functional categories containing more than one differentially expressed gene are connected by a shaded background to highlight functional differences in gene expression between temperature conditions (note that these connections are between the same functional categories, but different genes). Differential expression was defined by log2-fold change above 1 and an adjusted *P* value below 0.05.

*gapA* (Nonaka et al, 2006), are upregulated by the symbiont under cold conditions. The unusual circumstances in which we observe upregulation of *rpoH* and its regulon suggest that this alternative sigma factor responds to different stresses in the reed beetle symbiont than in its free-living relatives.

## Symbiont gene expression changes according to host needs across life stages

From larval to adult stage, reed beetles experience a change in their diet as well as the localization of the symbionts (Bieňkowski and Orlova-Bieňkowskaja, 2004; Buchner, 1965; Reis et al, 2020; Stammer, 1935). To understand how symbiont gene expression responds to changes in host needs across its life cycle, we assessed symbiont gene expression in four species of Donaciinae—*D. marginata*, *Donacia thalassina*, *Donacia versicolorea* and *Macroplea mutica*. The comparison across life stages for *D. marginata* was done using the warm acclimated larvae from the temperature

response experiment described above, which were processed without a cold incubation prior to dissection, while the adults of *D. marginata* and all other specimens used for this section were immobilized for 5 min at cold temperatures for dissection (see "Methods", Dataset EV3). However, as we observed similar patterns in gene expression across all four species (Fig. 2), the short cold exposure likely had no major effect on gene expression.

### The reed beetle symbiont upregulates amino acid and vitamin biosynthesis pathways during host larval stages

Out of the approximately 410 protein-coding genes in the symbiont genomes, we identified 57, 54, 34, and 27 genes differentially expressed between life stages for the symbionts of *D. marginata* (symbDMAR), *D. thalassina* (symbDTHA), *D. versicolorea* (symbDVER), and *M. mutica* (symbMMUT), respectively (Dataset EV4). Of these, 28/57, 18/54, 26/34, and 21/27 genes were upregulated during the host larval stage. During this stage, we observed an upregulation of amino acid biosynthesis pathways by

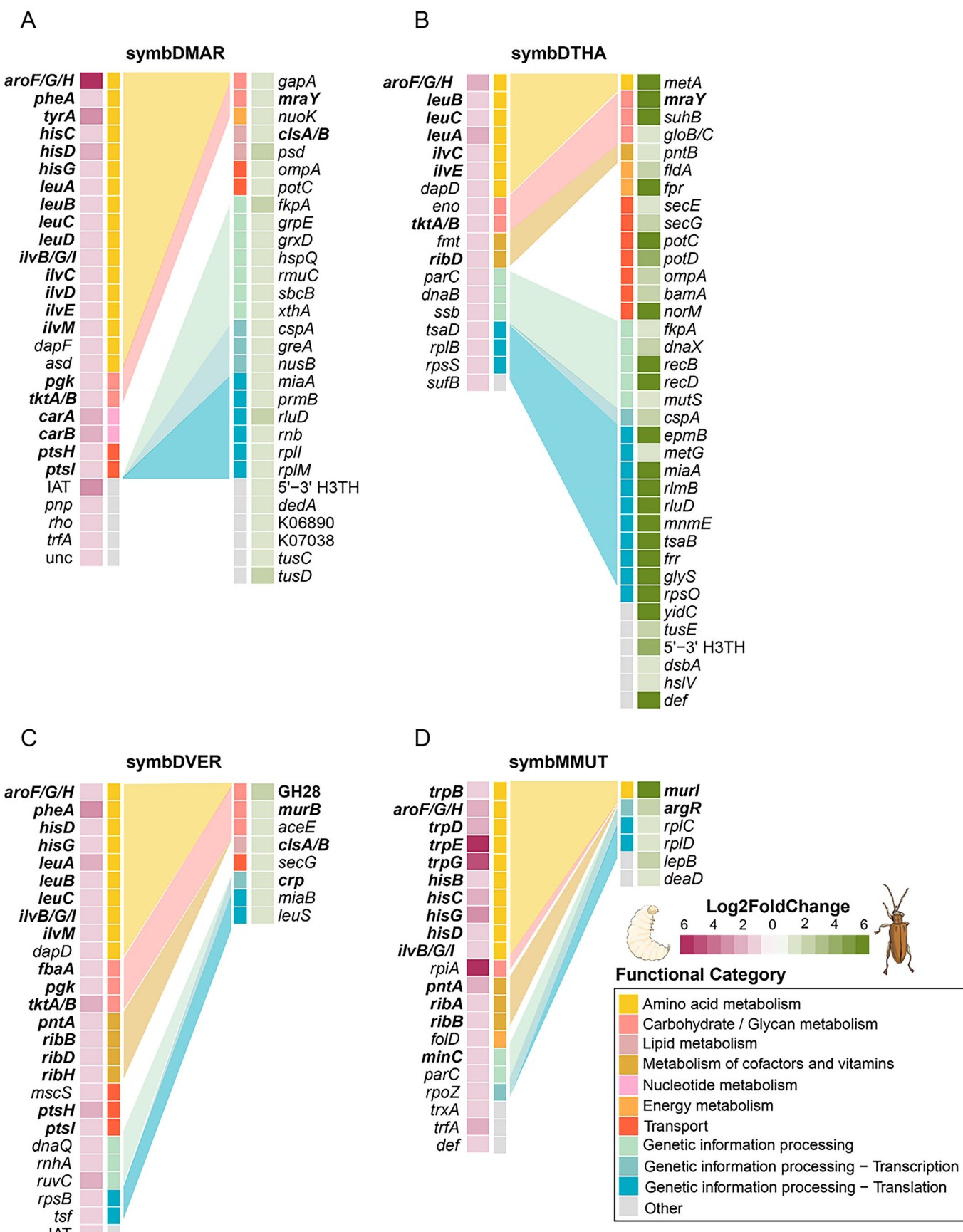

**Figure 2. Symbionts upregulate amino acid and carbohydrate metabolism genes in reed beetle larval stages.**

Displayed are significantly upregulated genes by the symbiont of *Donacia marginata* (**A**), *Donacia thalassina* (**B**), *Donacia versicolorea* (**C**), and *Macroplea mutica* (**D**) during larval (left) or adult (right) stages, respectively. Differential expression was defined by log2-fold change above |1| and an adjusted *P* value below 0.05 between the symbionts of larvae and adults within a species (Wald test with Benjamini–Hochberg correction for multiple testing). In the heatmap, different shades of magenta indicate genes upregulated in the larval stage, whereas green denotes genes upregulated in adults. Functional categories were manually curated after KEGG and InterProScan annotation. Functional categories containing more than one differentially expressed gene are connected by a shaded background to highlight functional differences in gene expression between host life stages.

the symbiont (Fig. 2), with 16/28, 8/18, 9/26, and 10/21 of genes upregulated in the larval stage by symbDMAR, symbDTHA, symbDVER and symbMMUT, respectively, being involved in amino acid metabolism. In particular, genes involved in the biosynthesis of aromatic amino acids (tyrosine, phenylalanine, and tryptophan), branched-chain amino acids (valine, leucine, isoleucine), and histidine were upregulated in larvae across several of the species, with *aroF* being the only gene in our analysis that was upregulated by all symbionts. An increase of AroF activity, together with *tkt* overexpression, is known to be an effective measure to increase tyrosine biosynthesis in *E. coli* (Draths et al, 1992; Yi et al, 2003). Across the three *Donacia* species, the symbiont upregulated *tkt* as well as other carbohydrate metabolism genes during larval stages (Fig. 2A–C), possibly increasing the availability of precursor substrates for the biosynthesis of aromatic amino acids (Lütke-Eversloh and Stephanopoulos, 2007). In fact, amino acid and carbohydrate metabolism can generally be coordinated via the phosphoenolpyruvate-carbohydrate phosphotransferase system (PTS). An accumulation of amino acid degradation products or tricarboxylic acid cycle intermediates inhibits the PTS in *E. coli*, resulting in reduced sugar intake and downregulation of amino acid biosynthesis (Doucette et al, 2011). Concordantly, during larval stages, symbDMAR and symbDVER significantly upregulated PTSI and PTSH, suggesting a higher sugar intake to fuel the upregulated amino acid biosynthesis. Accordingly, genes related to glycolysis/glucogenesis (*eno*, *fbaA*, *pgk*) and the pentose phosphate pathway (*fbaA*, *rpiA*, *tkt*) are upregulated by the symbionts during the larval stage of different hosts (Fig. 2).

In addition, symbMMUT showed a significant KEGG enrichment of histidine metabolism and aromatic amino acid pathway genes among the upregulated genes in larvae (Dataset EV2), including the tryptophan biosynthesis pathway (Fig. 2D). This pathway is unique to the symbiont of *M. mutica*, as the tryptophan biosynthesis genes have been lost in the ancestor of the *Donacia* symbionts after the split of the early diverging *Donacia tomentosa* (Reis et al, 2020). Across the symbionts of the three *Donacia* spp., genes involved in branched-chain amino acid biosynthesis were also consistently upregulated (Fig. 2A–C). The KEGG enrichment analysis showed that the genes upregulated by these three symbionts in the larval stage were significantly enriched in valine, leucine and isoleucine biosynthesis and 2-oxocarboxylic acid metabolism pathways (Dataset EV2). Conversely, only *ilvB/G/I* was upregulated in symbMMUT (Fig. 2D). Interestingly, symbMMUT has lost the final step in branched-chain amino acid biosynthesis from its genome, the branched-chain amino acid aminotransferase *ilvE* (Reis et al 2020). The final transaminase steps in the biosynthesis of branched-chain and aromatic amino acids are taken over by the host in several other symbiotic associations (Anbutsu et al, 2017; Hansen and Moran, 2011; Rabatel et al, 2013;

Russell et al, 2013; Shigenobu et al, 2000; Wilson et al, 2010), likely granting the host control over the flux through the biosynthetic pathway (Anbutsu et al, 2017; Rabatel et al, 2013). Thus, it seems possible that the *Donacia* symbionts are still capable of regulating branched-chain amino acid biosynthesis, while regulation has been outsourced to the host upon loss of *ilvE* in the symbiont of *M. mutica*. In addition, we observed an upregulation of several genes involved in other metabolic processes, such as carbohydrate metabolism and metabolism of cofactors and vitamins. Notably, genes involved in riboflavin biosynthesis (*ribB/D/H*) were significantly upregulated across the symbionts of all species except symbDMAR, and this pathway is enriched in the upregulated genes of symbDVER in larvae (Dataset EV2; Fig. 2).

In many symbiotic systems, the symbiont-provided benefits are regulated by adjusting the symbiont titers to the levels required to meet the host's nutritional demands (Whittle et al, 2021). Alternatively, regulation of gene expression in symbionts can also be achieved by gene transfer to plasmids (Latorre et al, 2005) and subsequent plasmid copy number variation (García-Lozano et al, 2024; Viñuelas et al, 2011). Previously, symbiont gene expression changes were shown in *Stammera*, which provides the same benefit throughout the host's life cycle, but experiences differential gene expression during extracellular transmission (García-Lozano et al, 2024). In our study, the upregulation of amino acid biosynthesis pathways by the symbionts during host larval stages suggests that the symbionts of the Donaciinae respond to differential host demands across life stages with changes in their transcriptional profiles. Some of the observed transcriptional changes can be linked to the TFs encoded in the symbiont's genome to derive hypotheses on the underlying gene regulatory pathways (Dataset EV5). The symbionts encode for four to five TFs: ArgR, c-AMP receptor protein (CRP), BolA, DksA, and fumarate and nitrate reduction regulatory protein (FNR—only present in pectinase-encoding symbionts), (Dataset EV5). Based on extensive work in model organisms such as *Escherichia coli*, to which the Donaciinae symbionts are closely related, these factors are known to respond to a variety of external stimuli, such as amino acid deprivation (Maas, 1994; Paul et al, 2005; Steinchen et al, 2020) and energy deficits (Gosset et al, 2004). As a response, the TFs are known to regulate the expression of amino acid biosynthesis genes (ArgR, DksA), general metabolic changes (CRP, RpoH), and changes in cell shape (BolA) in *E. coli*. Here, we derive hypotheses on the regulatory pathways by comparison of the predicted TFs, the upregulated genes, and the genomes of Donaciinae symbionts with the *E. coli* databases RegulonDB and EcoCyc (Dataset EV5). It needs to be stressed that our hypotheses on the possible roles of transcription factors in the Donaciinae symbionts are based on the gene expression profiles and still require experimental validation. The symbiont-encoded ArgR (N0.HOG0000347, Dataset EV6), a

repressor of arginine biosynthesis (Maas, 1994), could potentially regulate arginine biosynthesis. Contrary to other amino acid biosynthetic pathways encoded by the symbiont, the arginine biosynthesis pathway is incomplete, consisting only of *carA* and *carB*, which can synthesize carbamoyl phosphate either from ammonia or glutamine (Charlier et al, 2018), and *argF*, which can use the carbamoyl phosphate together with ornithine to produce

citrulline (Charlier et al, 2018). A similar scenario has been observed in *Blochmannia*, in which it was proposed that CarAB and ArgI synthesize citrulline to then be converted into arginine by the host (Williams and Wernegreen, 2010). An equivalent mechanism could be present in the symbiont of the Donaciinae, which might make this the only amino acid directly sensed and regulated by a TF in this bacterium.

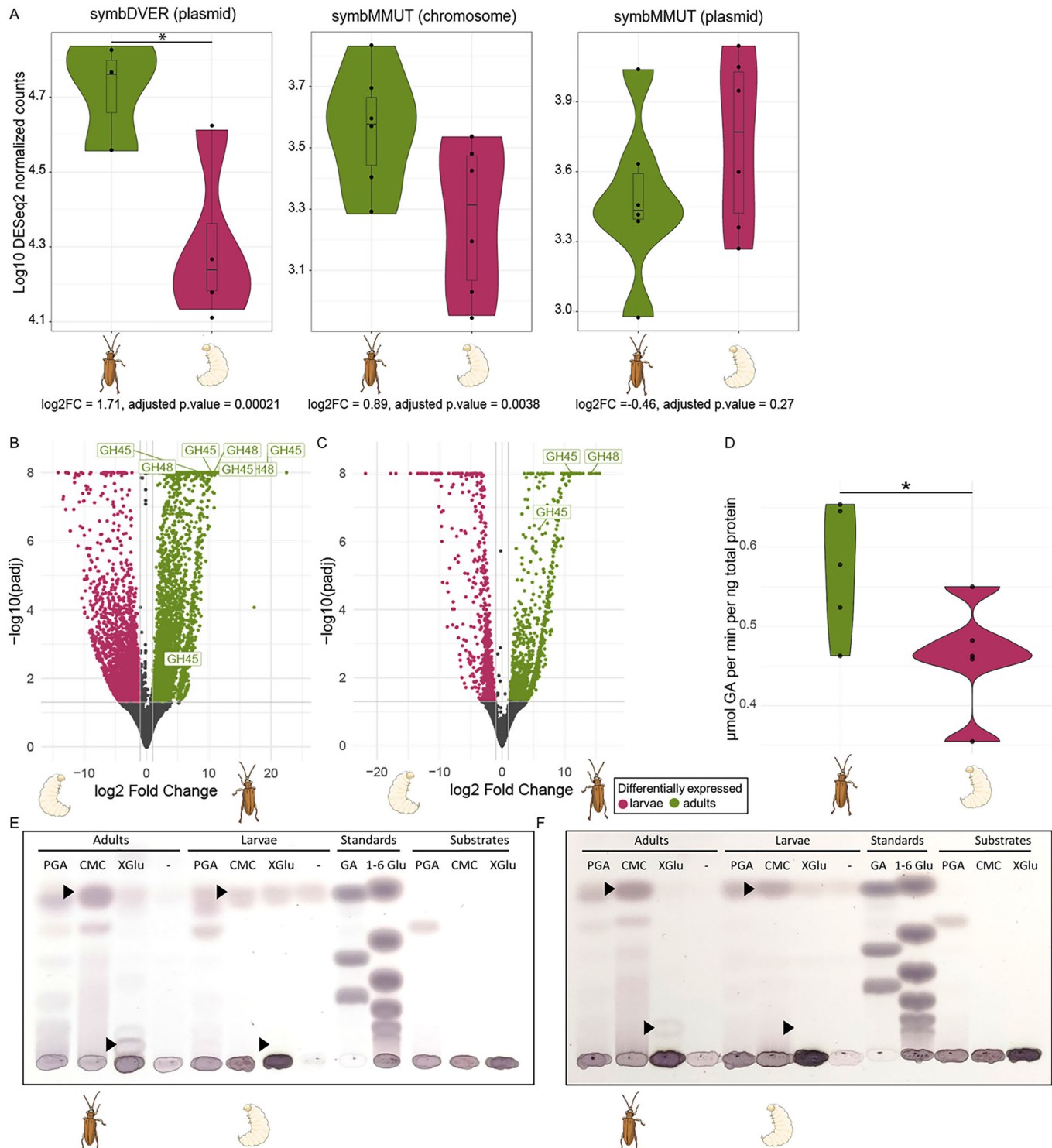

**Figure 3. Reed beetle adult guts show higher expression and activity of PCWDEs than larval guts.**

(A) Base 10 logarithm of the DESeq2 normalized counts for the symbiont-encoded pectinases on the plasmid of symbDVER (left; $n = 3$ adults, $n = 4$ larvae), the chromosome and the plasmid of symbMMUT (center and right; $n = 6$ adults, $n = 6$ larvae). Significant differential expression, indicated with an asterisk (Wald test with Benjamini–Hochberg correction for multiple testing), was identified by a log2-fold change above |1| and an adjusted $P$ value lower than 0.05. (B, C) Differential gene expression of beetle-encoded PCWDEs between adults and larvae of *D. versicolorea* (B) and *M. mutica* (C). Significantly differentially expressed genes were defined by log2-fold change above |2| and adjusted $P$ value below 0.05 (Wald test with Benjamini–Hochberg correction for multiple testing). (D) 3,5-dinitrosalicylic acid (DNS) assay for estimating quantitative enzymatic activity of *D. versicolorea* larvae and adults against polygalacturonic acid. All enzymatic assays were done with pools of three guts of adults and larvae of each species as replicates. For the DNS assay, 5 pools of 3 guts were used per stage. Statistical significance was determined with a two-sided $T$ test, $P$ value $= 0.0498$ and is indicated with an asterisk. (E, F) Enzymatic activity assays using guts of adults and larvae of *D. versicolorea* (E) and *M. mutica* (F). Arrows indicate presence or absence of relevant degradation products.

Most of the other amino acid biosynthesis pathways upregulated in the Donaciinae symbionts during the hosts' larval stages, such as the leucine, histidine, and tryptophan operons, are known to be regulated across Proteobacteria via a transcription attenuation mechanism (Vitreschak et al, 2006). Transcription attenuation takes place via changes in the conformation of the mRNA while it is being transcribed, induced by ribosome stalling (Vitreschak et al, 2006; Yanofsky, 1981). The stalling is caused by the low abundance of charged tRNAs during the translation of a leading peptide encoding for this amino acid (Vitreschak et al, 2006; Yanofsky, 1981), thus leading to the formation of an antitermination loop and the transcription of the whole operon (Vitreschak et al, 2006; Yanofsky, 1981). We speculate that the symbionts of the Donaciinae suffer from amino acid starvation during host larval stages, which results in low abundances of charged tRNAs inside the symbiont cell. This could then induce an increase in amino acid biosynthesis, as well as other metabolic responses, through transcription attenuation. Moreover, if the symbiont is able to synthesize ppGpp—an alarmone involved in response to starvation and other stresses both in bacteria and in plastids (Field, 2018; Mehrez et al, 2023; Sinha and Winther, 2021; Steinchen et al, 2020)—this molecule can bind to DksA or to the RNA polymerase directly and upregulate amino acid biosynthesis (Steinchen et al, 2020).

### Life-stage-specific expression of symbiont- and host-encoded PCWDEs

No symbiont genes were consistently upregulated in the adult stage across all four beetle species (Fig. 2), and no functional category was significantly enriched among the genes upregulated in the adult stage for any of the species (Dataset EV2). However, symbionts of adult beetles exhibited an overall larger proportion of upregulated genes related to genetic information processing (16/29, 18/29, 3/8, and 5/6 of upregulated genes compared to 2/28, 0/18, 0/26, and 3/21 genes upregulated during larval stages of symbDMAR, symbDTHA, symbDVER and symbMMUT, respectively). Given the folivorous feeding habit of adult beetles, we expected to observe an upregulation of the pectinases (GH28) by the symbiont during the adult stage of *D. versicolorea* and *M. mutica* (the symbionts of the other two host species do not encode pectinases, as outlined above). Upregulation of GH28 in the adult stage was indeed observed for *D. versicolorea* (Fig. 2C), but not for *M. mutica* (Fig. 2D). Contrary to symbDVER, symbMMUT encodes for two pectinases of different origins (Reis et al, 2020). In contrast to the significant upregulation of the plasmid-encoded symbDVER pectinase in adults, the plasmid-encoded pectinase in symbMMUT tended to have a higher expression in larvae than in adults,

although this difference was not significant (Fig. 3A). Conversely, the chromosome-encoded pectinase of symbMMUT was more highly expressed in adults compared to larvae with a significant adjusted $P$ value of 0.0038, but a log2-fold change of 0.89, which was below the cutoff of 1 used in this study (Fig. 3A). Regulation of the pectinase could be mediated by the putative FNR, a TF and the only gene whose presence perfectly correlates with the presence of pectinases across the symbiont genomes (see Supplementary Data in Reis et al, 2020). This correlation strongly indicates that this TF is involved in the regulation of GH28 or other processes related to this function. FNR belongs to the same TF superfamily as CRP, which has diversified its functions both regarding the regulated genes and binding molecules (Körner et al, 2003). It is thus possible that FNR has evolved to regulate GH28 or a related function in the symbionts of the Donaciinae.

The discrepancy in the expression of symbiont pectinases led us to further investigate whether the host only requires plant cell wall digestion during adulthood in both species, or whether larvae also rely on PCWDEs, possibly for accessing plant cell content in the roots. To assess plant cell wall digestion activity, we first performed RNA-seq of the guts of larvae and adults of both species. We analyzed the expression of relevant host-encoded PCWDE genes (McKenna et al, 2019; Shin et al, 2021), annotated with dbCAN, and observed that the expression of these genes was generally significantly higher in adult than larval guts in both *D. versicolorea* (Fig. 3B) and *M. mutica* (Fig. 3C; Dataset EV7). Adults of both species upregulated GH45 and GH48, predicted to be cellulases and xyloglucanases, respectively (Fig. 3B,C). To corroborate our findings on the protein activity level, we extracted the proteins from three guts of larvae and adults of *D. versicolorea* and *M. mutica*, respectively, and tested for the enzymatic activity of the gut extracts against standard polymers analogous to common plant cell wall polymers, namely the pectin backbone polygalacturonic acid (PGA), carboxymethyl cellulose (CMC), and xyloglucan (XyGlu) using thin-layer chromatography (TLC) (Fig. 3E,F). For both species, the gut extracts of larvae contained carbohydrate monomers in the absence of any added substrate (Fig. 3E,F), leading to the presence of these products throughout the TLC plate. In accordance to the transcriptomic data from the host's guts in both species, we verified that the plant cell wall digestive activity is largely carried out in adult guts as indicated by the higher accumulation of monomers and oligomers in the reaction of the substrates with adult gut extracts compared to the one observed in larvae. Specifically, for cellulose (represented by CMC) the band corresponding to glucose was more intense in adults compared to larvae, indicating a more complete digestion. In the case of xyloglucan, degradation products were only detectable in adults

and not in larvae. Altogether, these results support the digestion of PCW material taking place mostly during adult stages. Due to several breakdown products represented by multiple bands on TLC, it is difficult to assess the pectinase efficiency across life stages. Thus, we quantified pectinase activity across the life stages of *Donacia versicolorea* with a quantitative 3,5-dinitrosalicylic acid (DNS) assay, while we unfortunately did not have enough material of the rare species *M. mutica* available for a DNS assay. However, the assay confirmed that, on average, *D. versicolorea* adults showed significantly higher enzymatic activity against PGA than larvae (Fig. 3D; Dataset EV8). Thus, we show that in *Donacia versicolorea* the expression of the symbiont-encoded pectinase is concordant with the breakdown of pectin and upregulation of PCWDEs preferentially in adult beetles.

## sRNA as a potential regulator of symbiont gene expression in the reed beetle symbiont

In other symbionts with a reduced genome such as the aphid endosymbiont *Buchnera*, regulation of symbiont gene expression across host life stages can also be achieved by small non-coding RNAs (sRNAs) (Hansen and Degnan, 2014; Thairu et al, 2018; Thairu and Hansen, 2019). sRNAs have been implicated in the regulation of several amino acid biosynthesis pathways. In fact, an antisense sRNA was validated in affecting the translation of *carAB* (Thairu et al, 2018), and several others likely target the branched-chain amino acid biosynthesis, aromatic amino acid biosynthesis, the pentose phosphate pathway, and riboflavin biosynthesis in *Buchnera* (Thairu and Hansen, 2019). We searched the transcriptomic data for each life stage of each Donaciinae species for putative symbiont sRNAs using APERO (Leonard et al, 2019), by investigating transcripts below 500 nucleotides in length that were classified as either antisense to a coding sequence or located in an intergenic region. By combining the results for both life stages for each species, we obtained 154 to 364 putative sRNA regions. We then analyzed the life-stage-specific expression patterns of all putative sRNA, revealing 6, 76, 9, and 11 differentially expressed sRNAs for *D. marginata*, *D. thalassina*, *D. versicolorea*, and *M. mutica*, respectively, which were then analyzed in terms of predicted free energy and genomic localization. To identify an sRNA candidate, we expected a predicted minimum free energy secondary structure below the first quantile of the free energy distribution of 100 randomized sequences of the same length and GC content. Additionally, the same candidate sRNA should be predicted and differentially expressed in the same direction in more than one species. However, none of the putative sRNAs met these criteria (Datasets EV9 and EV10). Thus, while sRNAs possibly play a role in the regulation of symbiont processes, we could not identify any putative regulatory sRNAs that were shared among several of the symbionts. However, some of the putative sRNAs identified with APERO were antisense to the same genes or pathways as the ones found in *Buchnera*. Namely, an sRNA antisense to *carB* was detected in symbMMUT, but this gene was not differentially expressed (Datasets EV9 and EV11). The same was true for an sRNA antisense to the 3'-region of *prmB* and 5'-region of *aroC* in symbDTHA, and a putative sRNA antisense to the 5'-region of *aroC* in symbMMUT (Dataset EV11). In addition, putative sRNAs were identified antisense to the 5'-region of the transketolase gene (*tkt*) across all symbionts (Dataset EV11). Thus, although there is

potential for sRNAs to have a similar regulatory role as in *Buchnera*, further studies pairing transcriptomics focused on small transcripts and proteomics will be necessary to better understand the potential role of sRNA-mediated regulation of gene expression in the symbionts of the Donaciinae.

## Symbiont cell shape changes across host life stages

The symbionts of the Donaciinae encode genes for synthesizing their own cell wall and likely regulate their own cell division, as they encode genes for peptidoglycan, lipid and cardiolipin biosynthesis, and the operons *min*CDE and *mre*BCD (Reis et al, 2020), which regulate cell division (Hu and Lutkenhaus, 1999) and shape in *E. coli* (Wachi et al, 1989), respectively. The symbionts of all of the four investigated Donaciinae species showed differential expression of some of the genes putatively involved in the regulation of cell shape across host life stages, but the expression was not uniform across species (Fig. 2). This was observed for peptidoglycan biosynthesis genes during adult stages for symbD-MAR and symbDTHA (*mraY*), symbDVER, and symbMMUT (*murB* and *murI*, respectively). Additionally, symbDMAR and symbDVER upregulated their cardiolipin synthase gene (*cls*) in adult beetles. Conversely, symbMMUT upregulated the Z-ring positioning protein *minC* during host larval stages (Fig. 2). Despite the diversity of processes, it seems that, in general, there is higher peptidoglycan metabolism taking place during host adult stages. Peptidoglycan biosynthesis is required not only for elongation of the cells, but also for septation (Navarro et al, 2022). In symbDMAR and symbDVER, upregulation of cardiolipin during host adult stages could point to a role of this lipid in a more spherical shape. Cardiolipin is associated with curvature (Beltrán-Heredia et al, 2019) and accumulates in smaller cells (Koppelman et al, 2001). Moreover, upregulation of *minC* during host larval stages by symbMMUT could reflect an inhibition of cell division (Rowlett and Margolin, 2013).

Using fluorescence in situ hybridization (FISH), we assessed whether the differences in gene expression corresponded to changes in symbiont cell shape in the symbiotic organs of larvae and adults of *D. thalassina* and *M. mutica*, respectively. We observed that the symbionts of both species were indeed elongated and enlarged in larvae (Fig. 4A,C) compared to adults (Fig. 4B,D). In *D. thalassina* larvae, the symbionts showed a filamentous morphology, whereas they acquired a variety of irregular shapes in *M. mutica* larvae (Fig. 4C). In the adults of both species, however, the symbionts showed a regular round shape in the symbiotic organs (Fig. 4B,D) that was consistent across replicates (Fig. EV1). Descriptions of differences in cell shape of the symbiont across life stages have been previously published for *Donacia semicuprea* (Stammer, 1935). Although there are several instances of symbionts with different cell shapes across host life stages and/or localizations, the role and regulation of this phenotype remain unclear.

Changes in symbiont cell shape throughout the host life cycle or for transmission are common across insect symbioses (Buchner, 1965, pp. 187, 210; Wang et al, 2023)—and the regulation is often attributed to the host. In the grain pest beetle *Oryzaephilus surinamensis* (Silvanidae), the bacterial symbionts change their cell shape throughout the host's life cycle (Buchner, 1965) without encoding for the machinery that would allow for it to synthesize its own cell envelope or regulate cell shape (Engl et al, 2018). In the aphid symbiont *Buchnera*,

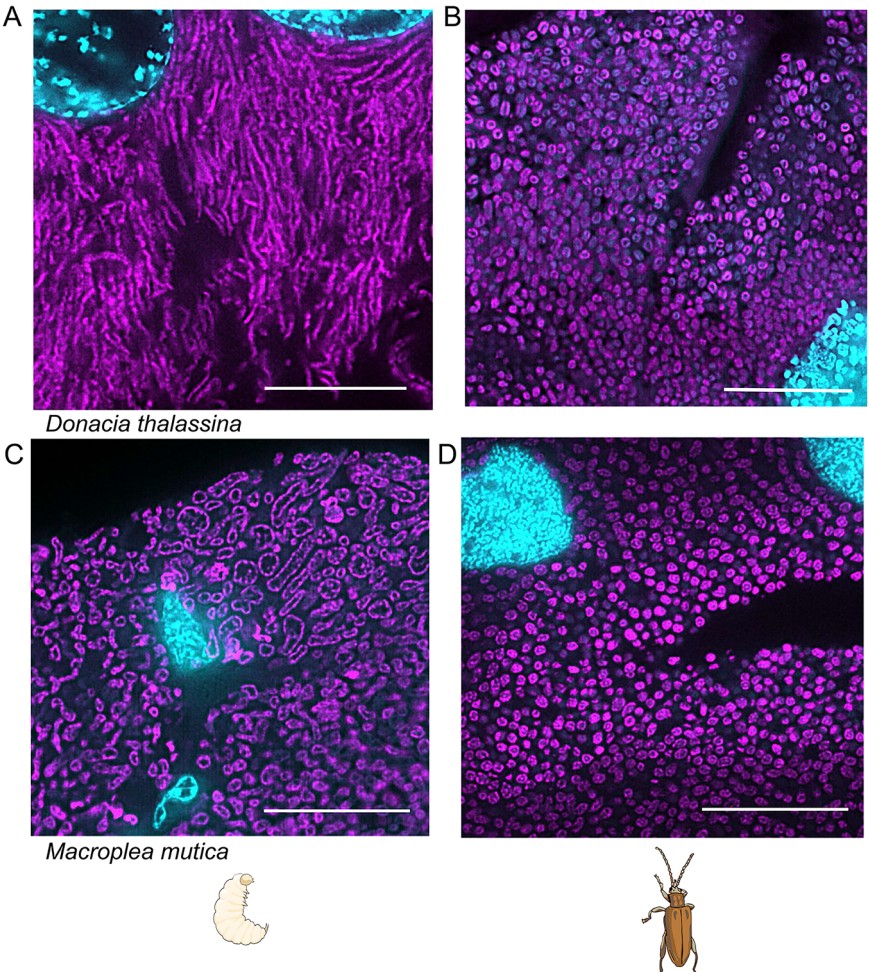

**Figure 4. Symbiont morphology changes between larval and adult life stage.**

Symbionts in the symbiotic organs of a larva (**A**) and an adult (**B**) *D. thalassina*, and a larva (**C**) and an adult (**D**) *M. mutica*. Overlay of a probe specific to Donaciinae symbionts (magenta) and DNA staining with DAPI (cyan). Scale bars correspond to 20 μm.

the machinery for biosynthesis and regulation of the bacterial cell wall consists of a patchwork of symbiont- and host-encoded genes, with the latter deriving from horizontal gene transfer events from different bacteria (Smith et al, 2022). Even when the symbiont retains the full capability for making and regulating its own cell wall, such as in the *Sodalis-Sitophilus* symbiosis, the symbiont cell shape is affected by the host via an antimicrobial peptide, ColA (Login et al, 2011). In other systems, cell envelope biosynthesis pathways are still present in the symbiont genome, but little is known about the mechanisms regulating cell shape and division in symbionts with highly eroded genomes. In the reed beetles, the symbiont encodes for BolA, which has been described in *E. coli* as a regulator of cell shape upon entry into the stationary phase and response to stress (da Silva et al, 2023).

The conservation of cell shape changes across symbionts with divergent genome sizes suggests that there is an adaptive value of such changes in insect-associated symbionts. The localization in different symbiotic organs across the host's life cycle, like in the Donaciinae symbiosis, may correspond to specific nutritional or physicochemical environments that could favor different symbiont cell shapes. In

addition, several beneficial symbionts present smaller, rounder cells in transmission organs and/or eggs (Buchner, 1965; Stammer, 1935), and some pathogens also adopt a smaller shape upon transmission (Yang et al, 2016). Transmission can be a challenging process for a bacterium, both in the case of pathogens and mutualists. Transmission of mostly intracellular symbionts can require periods of extracellularity and even outside the host's body (Buchner, 1965; Oguchi et al, 2024; Reis et al, 2020). Upon transmission, the host may restrict the nutrients available to the symbiont, as keeping the symbiont is costly (Vigneron et al, 2014), or the symbiont may be isolated in extracellular structures such as caplets or capsules (Pons et al, 2022; Stammer, 1935). Smaller, rounder shapes can be advantageous to survive these harsher conditions. In fact, *E. coli* and *Salmonella enterica* respond to nutrient starvation by reducing their size and elongate again upon nutrient provisioning (Roszak and Colwell, 1987). In addition, colonization and, consequently, successful transmission to the next generation may be facilitated by a smaller cell, which may be easier to transmit and uptake (Yang et al, 2016). While we observe changes in cell morphology in the genome-reduced Donaciinae symbionts and

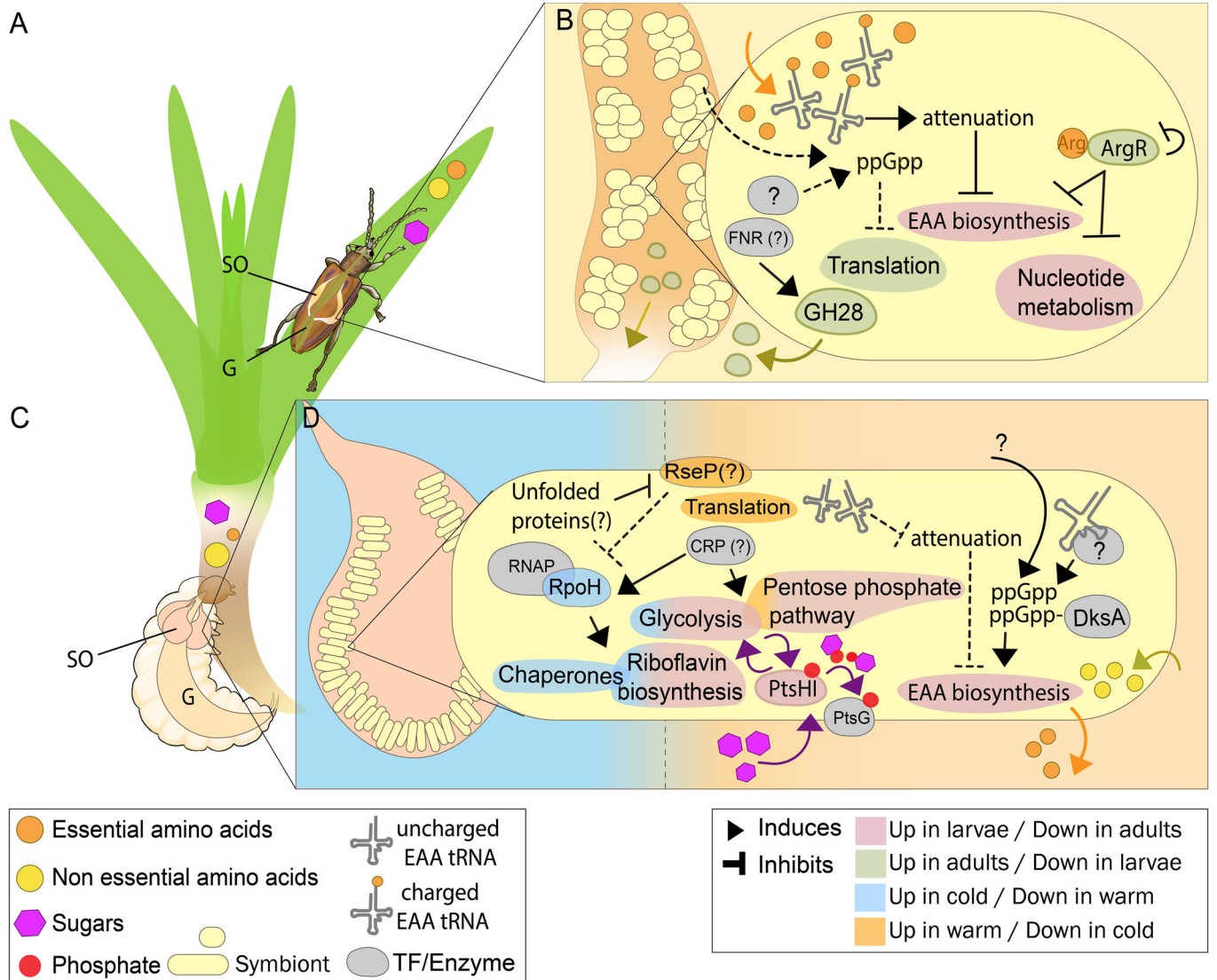

**Figure 5. Schematic overview of putative symbiont gene regulatory mechanisms during host larval and adult stages, based on gene expression profiles and predicted TF functions.**

(A) The adults feed on leaves of their host plant (Wachi et al, 1989). (B) In the small and round symbiont cells, pectinase is upregulated, possibly by FNR, and transported from the symbiotic organs into the midgut. The abundance of essential amino acids obtained from the host diet, bound to tRNAs, promote attenuation and potentially prevent ppGpp synthesis, leading to the repression of EAA biosynthesis. Arginine binds to the arginine repressor, ArgR, which inhibits the synthesis of arginine precursors or nucleotides. (C) The larvae feed on the sap from their host plant's roots, a diet rich in carbon but poor in nitrogen. (D) At cold temperatures, unfolded proteins potentially accumulate, triggering upregulation of *rpoH* and its regulon, including the expression of chaperones. At warmer temperatures, the elongated or irregularly shaped symbionts increase their translation activity. The scarcity of charged tRNAs for EAAs potentially triggers the production of ppGpp and prevents attenuation of amino acid biosynthesis pathways. Sugars are imported and fueled into the glycolytic and pentose phosphate pathways. SO symbiotic organs, G gut.

identify certain genes potentially involved in the regulation of cell shape, further experiments are needed to demonstrate the regulatory roles of the TFs in the Donaciinae symbionts and assess the adaptive value of cell shape changes across the host life stages.

# Conclusions

Here we show that the symbionts of Donaciinae, despite having severely eroded genomes of around 0.5 Mbp, exhibit specific transcriptional responses that allow them to react to changes both within and across host life stages. Furthermore, we connect these changes to the encoded transcription factors, based on predictions of TF functions from studies on model organisms. The symbionts potentially use their alternative sigma factor, RpoH, at lower temperatures to increase chaperone expression (Fig. 5). Moreover, in the four examined species, the Donaciinae symbionts show a consistent response at the transcriptional level to the changing needs of the host across its life stages. These mechanisms enable the biosynthesis of essential amino acids to be regulated by the

symbiont and respond to a demand for amino acids in the larvae (Fig. 5). Furthermore, we observed that the symbiont-encoded pectinase is significantly upregulated during adult stages in only one of the two species where it is present. However, we found that activity and upregulation of host-encoded PCWDEs is consistent in the adults of both species, supporting the notion that PCWDEs are particularly required in the folivorous adults to break down the PCW. Additionally, the Donaciinae symbionts differentially express individual genes related to bacterial cell shape and cell envelope biosynthesis in the four reed beetle species we investigated. This corresponds to host life-stage-specific differences in cell shape that had been previously reported for the symbiont of *Donacia semicuprea* and that we also observe for the symbionts of *D. thalassina* and *M. mutica*. Our observations on gene expression patterns across host life stages allow us to propose putative roles for the limited repertoire of TFs encoded in the symbionts' genomes, which, however, still require experimental validation in tractable systems: While ArgR, CRP, CsrA, and DksA are likely involved in regulating amino acid biosynthesis and carbon metabolism, FNR may regulate pectinase expression or related functions, and BolA is potentially involved in the cell shape change (Fig. 5). These TFs encoded by the Donaciinae symbionts are also found in other symbionts with both larger and smaller genome sizes (Dataset EV12), but their roles so far remained largely unknown. Our work suggests that, together with TF-independent mechanisms such as transcription attenuation, these TFs may play a pivotal role in responding to the abiotic environment and to the host's physiological needs. These findings open the door to a more comprehensive understanding of the gene regulatory potential in long-term host-associated symbionts approaching a minimal gene set.

# Methods

### Reagents and tools table

| Reagent/resource | Reference or source | Identifier or catalog number |
| --- | --- | --- |
| **Experimental models** | | |
| Donaciinae adult beetles and larvae | This study, see Dataset EV11 | |
| **Recombinant DNA** | | |
| NA | NA | NA |
| **Antibodies** | | |
| NA | NA | NA |
| **Oligonucleotides and other sequence-based reagents** | | |
| PCR and sequencing primers | This study; Folmer et al, 1994 | NA |
| **Chemicals, enzymes, and other reagents** | | |
| OneTaq2x Master Mix | New England Biolabs | M0482S |
| Laemmli buffer | BioRad | #1610737 |
| beta-mercaptoethanol | Applichem | A1108 |
| Tris/Glycine/SDS buffer | BioRad | |
| Page Blue Protein Stain | Thermo Fisher | 24620 |

| Reagent/resource | Reference or source | Identifier or catalog number |
| --- | --- | --- |
| Protein assay Dye Concentrate | BioRad | #5000006 |
| Albumin fraction V | Roth | 8076.2 |
| Polygalacturonic Acid from Citrus Pectin | Megazyme | |
| Carboxymethyl Cellulose 4M | Megazyme | |
| Xyloglucan from tamarind seed | Megazyme | |
| Silica gel 60, 20 ×20 cm | Merck | |
| Formaldehyde 37% | Acros | |
| tert-Butanol | Roth | AE16 |
| Technovit 8100 | Kulzer GmbH | |
| DAPI | Roth | 6335.1 |
| SDS | Roth | 4360 |
| Tris | Roth | 4855 |
| Tris-HCl | Roth | 9090 |
| NaCl | Roth | 9265 |
| EtOH | Roth | 9065 |
| EDTA | Roth | 8040 |
| ProLong Diamond Antifade Mountant | Thermo Fisher | P36965 |
| VECTASHIELD Antifade Mounting Medium | Biozol | H-1000-10 |
| **Software** | | |
| Trim Galore v0.6.10 | Krueger et al, 2023 | |
| Sortmerna v4.3.6 | Kopylova et al, 2012 | |
| TopHat v2.1.1 | Kim et al, 2013 | |
| Bowtie2 v2.2.5.0 | Langmead and Salzberg, 2012 | |
| Samtools v1.13 | Danecek et al, 2021 | |
| FADU v1.8.3 | Chung et al, 2021 | |
| rnaSPADES v3.15.5 | Bushmanova et al, 2019 | |
| CDHIT v4.8.1 | Fu et al, 2012; Huang et al, 2010; Li et al, 2001, 2002, 2012; Li and Godzik, 2006; Niu et al, 2010 | |
| BlobTools v1.1 | Laetsch and Blaxter, 2017 | |
| dbCAN v4.1.2 | Yin et al, 2012 | |
| HMMER | http://hmmer.org/ | |
| KEGG | Kanehisa and Goto, 2000 | |
| KEGGREST | Tenenbaum and Maintainer, 2024 | |
| Interproscan v5.62 | Jones et al, 2014; Paysan-Lafosse et al, 2023 | |
| OrthoFinder v2.5.5 | Emms and Kelly, 2019 | |
| R v4.3.3 | Ripley, 2001 | |
| DESeq2 v1.40.2 | Love et al, 2014 | |
| clusterProfiler v4.8.3 | Wu et al, 2021 | |

| Reagent/resource | Reference or source | Identifier or catalog number |
|---|---|---|
| AnnotationDbi v1.62.2 | http://bioconductor.org/packages/AnnotationDbi/ | |
| APERO v1.0.3 | Leonard et al, 2019 | |
| GFFcompare v0.12.6 | Pertea and Pertea, 2020 | |
| featureCounts v2.0.1 | Liao et al, 2014 | |
| Mauve v1.1.3 | Darling et al, 2004 | |
| RNAfold v2.4.14 | Lorenz et al, 2011 | |
| Leica Application Suite X software | Leica | |
| FIJI | Schindelin et al, 2012 | |
| **Other** | | |
| Quick-DNA Tissue/Insect Microprep Kit | Zymo | D6015 |
| DNA Clean & Concentrator Kit | Zymo | D4004 |
| 3730xl DNA Analyzer | Thermo Fisher | A41046 |
| innuPrep DNA/RNA Mini kit | Analytik Jena | 845-KS-2080050 |
| TURBO DNA-free™ Kit | Invitrogen | AM1907 |
| Agilent Bioanalyser | Agilent | |
| Pan-Bacteria riboPOOL | siTOOLs Biotech | dp-P012-26 |
| NEBNext® Ultra™ II Directional RNA Library Prep Kit | New England Biolabs | E7760L |
| NextSeq 2000 | Illumina | |
| TissueLyser LT | Qiagen | 85600 |
| Zeba Desalt Spin Columns | Thermo Fisher | A57760 |
| Criterion XT precast gel 4-12% | BioRad | 3450123 |
| Criterion Cell | BioRad | 4206277 |
| Azure Biosystems | Biozym | |
| Infinite M200 plate reader | Tecan | |
| Leica RM 2245 microtome | Leica | |
| Leica THUNDER imager Cell Culture 3D with a K5 monochrome camera | Leica | |

## Insect collection, rearing, dissection, and barcoding

All insects were collected in Germany. Most adult beetles were kept at RT (22–23 °C) unless indicated otherwise (Dataset EV3). Larvae were kept in soil/debris completely submerged in tap water with soil and roots from either field-collected or greenhouse-reared plants (Dataset EV3). *Donacia marginata* larvae were reared from eggs laid in Aug/Sep 2021 on greenhouse-grown *Sparganium erectum* plants. For the assessment of symbiont transcriptomic response to temperature, plants with *D. marginata* larvae were split between two treatments simulating winter and summer conditions: 12/8 °C with an 8/16 h day/night temperature cycle, and 22 °C/14 °C with an 16/8 h day/night temperature cycle. The larvae were kept in constant darkness, as they naturally inhabit the sediment of lakes, ponds or small streams.

## Identification of beetle species

Adult reed beetles were identified to species level morphologically, whereas molecular barcoding was used to identify larvae, as they cannot be distinguished morphologically. For barcoding, the cytochrome oxidase I gene (COI) was amplified with the primers derived from Folmer et al (Folmer et al, 1994): HCO2198 and dgLCO1490 (Table EV1) to barcode larvae from places with co-occurring species for transcriptomics, imaging and DNS enzymatic assays. Since amplification efficiency was mixed, especially for *D. versicolorea* samples, a second primer pair was designed to amplify the Donaciinae COI. As such, for larvae used for the thin-layer chromatography (TLC) enzymatic assays, the primers LCO1490_Dver and HCO2199_Dver (Table EV1) were used. DNA was extracted using Quick-DNA Tissue/Insect Microprep Kit from Zymo (Irvine, CA, USA) according to the manufacturer's protocol. The PCR was done using OneTaq2x Master Mix (NEB, Ipswich, MA, USA) and the product was purified using DNA Clean & Concentrator Kit (Zymo). The purified product was sequenced on a 3730xl DNA Analyzer (Thermo Fisher Scientific, Waltham, MA, USA). The output sequences were then aligned to the mitochondrial genomes of Donaciinae species for species identification (Reis et al, 2020).

### Dissection and sample storage

Adult beetles were immobilized at −20 °C for 5 min prior to dissection. The larvae were placed in ice-cold water for 5 min prior to dissection, with the exception of the larvae used in the warm acclimation, which were kept at room temperature. The dissections were performed using PBS. The symbiotic organs of each individual were pooled together. The symbiotic organs and gut samples were immediately frozen in liquid nitrogen and subsequently stored at −80 °C for transcriptomics and enzymatic assays. Larval symbiotic organs for imaging were immediately transferred to fixative (see below). Information on the number of biological replicates used across the transcriptomics experiment can be found in Table 1, each replicate corresponds to the pooled symbiotic organs of a single individual or the gut of an individual.

## DNA and RNA extraction

RNA and DNA was extracted using the innuPrep DNA/RNA Mini kit (Analytik Jena, Jena, Germany) according to the manufacturer's protocol, following homogenization with metal beads for 3 min at 50 Hz in 450 µl of Lysis Solution RL. The samples were eluted in 30 µl of milliQ water. An additional DNase treatment was done immediately to the RNA samples using the TurboDNase kit (Invitrogen, Carlsbad, CA, USA) according to the manufacturer's protocol. Samples were kept at −80 °C after extraction.

## RNA library preparation and sequencing

RNA sequencing was performed at the Max-Planck Genome Center (Cologne, Germany). RNA was quality assessed with Agilent Bioanalyser, and then rRNAs were removed by a mixture of 50% of oligonucleotides designed against rRNAs of *Aedes albopictus* and 50% of oligonucleotides of the Pan-Bacteria riboPOOL kit, both from siTools, according to the protocol of the vendor. Depleted RNA was then processed with NEBNext® Ultra™ II Directional

**Table 1. Replicates used for the transcriptomics experiments**

| | Target tissue | Species | No. of larva replicates | No. of adult replicates |
|---|---|---|---|---|
| Across temperatures | Symbiotic organs | *Donacia marginata* | 2 (cold) and 3 (warm) | NA |
| Across life stages | | *Donacia marginata* | 3 (warm) | 3 |
| | | *Donacia thalassina* | 3 | 3 |
| | | *Donacia versicolorea* | 4 | 3 |
| | | *Macroplea mutica* | 6 | 6 |
| | Guts | *Donacia versicolorea* | 4 | 3 |
| | | *Macroplea mutica* | 3 | 3 |

RNA Library Prep Kit for Illumina® to achieve an Illumina-compatible library, followed by paired-end sequencing with 2 ×150 bp on a NextSeq 2000 device with a depth of ~25,000,000 reads per library.

## RNA-seq analysis

### Read processing

All reads were trimmed using Trim Galore (Krueger et al, 2023) v0.6.10 for a Phred quality score threshold of 30 and minimum read length of 20 bp. Potential remaining rRNA reads were removed using Sortmerna (Kopylova et al, 2012) v4.3.6 against the default database smr_v4.3_default_db with the rRNA of the symbionts of the Donaciinae added to it. Reads that did not map to rRNA were then used in the subsequent analysis.

### Symbiont read mapping and counting

Reads were mapped against the complete genomes (including the plasmids) of symbionts of *Donacia marginata*, *Donacia thalassina*, *Donacia versicolorea* or *Macroplea mutica* (accessions GCF_012567685.1, GCF_012568245.1, GCF_012570965.1 and GCF_012571345.1, respectively), respectively, using TopHat (Kim et al, 2013) v2.1.1 with Bowtie2 (Langmead and Salzberg, 2012) v2.2.5.0. Mapping was done in sensitive mode from end to end allowing for no discordant and no mixed reads for forward-reverse first strand stranded library. The BAM files were sorted and indexed using Samtools (Danecek et al, 2021) version 1.13. The aligned reads were counted with FADU (Chung et al, 2021) v1.8.3 using the GFF files from the above-mentioned genomes for CDS features.

### Host gut transcriptome assembly, mapping, counting, and functional annotation

Reads of adult and larval guts of each species were merged and used to assemble the gut transcriptomes of *Donacia versicolorea* and *Macroplea mutica* using rnaSPADES (Bushmanova et al, 2019) v3.15.5. A total of 562,977 and 181,100 contigs were obtained, respectively. The transcriptomes were then collapsed using CDHIT with default parameters (0.9 identity threshold and 10 as word length) (Fu et al, 2012; Huang et al, 2010; Li et al, 2001, 2002, 2012; Li and Godzik, 2006; Niu et al, 2010) v4.8.1, leaving 495,835 and 171,288 contigs. The reads of each sample were then mapped to the respective transcriptome using TopHat as described above. The reads mapped to each transcript were counted using Salmon v0.14.1. The relative contig coverage, GC content and contig

taxonomic classification were scanned after transcriptome assembly using BlobTools (v1.1), searching against UniProt and NR databases (Laetsch and Blaxter, 2017) (Dataset EV13) to enable the identification and removal of potential microbial symbionts or contaminants. Transcripts assigned to chloroplasts, bacteria, archaea or fungi were removed from the analysis. Carbohydrate digestive enzymes were annotated using dbCAN (Yin et al, 2012) v4.1.2. Only the enzymes identified with HMMER (HMMER, 2025) were included in subsequent analysis.

### Symbiont genome functional annotation

The translated CDS were used to obtain the KEGG (Kanehisa and Goto, 2000) functional annotation. Annotation levels were obtained using the R package KEGGREST (Tenenbaum and Maintainer, 2024) (Dataset EV6). Insight into ambiguous annotations or hypothetical genes was obtained by using Interproscan v5.62 (Jones et al, 2014; Paysan-Lafosse et al, 2023) (Dataset EV1) and the NCBI annotations. Orthogroups were defined by OrthoFinder (Emms and Kelly, 2019) v2.5.5 with 98.8% of total genes assigned to 393 orthogroups (Dataset EV6). Of these, 360 orthogroups had all species present, and 339 of these consisted only of single-copy genes, with 15 orthogroups containing two genes per species and 33 containing less than one gene per species.

### Differential gene expression analysis

Differential gene expression was assessed using the DESeq2 package v1.40.2 (Love et al, 2014) in R (Ripley, 2001) v4.3.3. Genes or transcripts with less than a total of 20 reads across samples were excluded from the analysis. In the host gut transcriptomics, this left 99,600 and 16,187 transcripts for *Donacia versicolorea* and *Macroplea mutica*, respectively. For all differential gene expression, the function lfcshrink was run using the adaptive Student's t prior shrinkage estimator from the package apeglm (Zhu et al, 2019). Genes were considered differentially expressed when the log2FoldChange was equal or above |1| for symbiont genes or |2| for host genes, and the adjusted $P$ value was lower than 0.05 (Datasets EV1, EV4, EV7, and EV9). The $P$ value was determined using the Wald test, the adjusted $P$ value was obtained using Benjamini–Hochberg (Benjamini and Hochberg, 1995) false discovery rate for multiple test correction.

### KEGG enrichment analysis

Enrichment of KEGG pathways in the genes differentially expressed at each life stage in each species was tested using the package clusterProfiler v4.8.3 (Wu et al, 2021) in R version v4.3.3 (Ripley,

2001). To do so, gene names obtained from the KEGG functional annotations were mapped to their respective Entrez ID using the R package AnnotationDbi v1.62.2 (AnnotationDbi, 2025) to the *E. coli* K12 reference in org.EcK12.eg.db v3.17.0 (96.63, 95.38, 94.79 and 94.6% of genes in symbDMAR, symbDTHA, symbDVER and symbMMUT mapped successfully). The enrichKEGG function was run against the KEGG "eco" database with the *P* value adjusted with the Benjamini and Hochberg method (Benjamini and Hochberg, 1995) and *P* value cutoff of 0.05 for gene sets ranging from 2 to 500 genes and a *q* value cutoff of 0.2 (Dataset EV2).

### sRNA identification

Putative sRNAs were predicted using APERO v1.0.3 (Leonard et al, 2019) according to Leonard et al 2019 (maximum width and distance = 10, enrichment = 0.05, minimum read number = 10) in R v4.4.2. We analyzed the bam files merged for each life stage per species using Samtools (Danecek et al, 2021) (Dataset EV11). Potential sRNAs below 500 nucleotides and classified as antisense or intergenic were kept in the output of adults and larvae for each species. The results for the different life stages for each species was merged using GFFcompare v0.12.6 (Pertea and Pertea, 2020) and boundaries for potential sRNA regions were determined based on the loci annotation (one locus = one potential sRNA region) (Dataset EV14). The resulting GTF was then used together with the individual bam file for each replicate to count reads mapping to each locus using featureCounts v2.0.1 (Liao et al, 2014). We then performed differential gene expression analysis with DESeq2 between the life stages of each species as described above (Dataset EV9). The position of the potential sRNAs that were differentially expressed in each species relative to each other was observed using a Mauve v1.1.3 whole genome alignment (Darling et al, 2004). Similar to what was done in Hansen and Degnan, 2014 and Thairu and Hansen, 2019, we compared the predicted free energy of the secondary structure of differentially expressed putative sRNA regions with the distribution of the free energy predicted for 100 replicates of randomized sequences of the same GC content and length. To do so, we used RNAfold v2.4.14. Similarly to the aforementioned publications, we defined that sRNAs of interest would be present in the same genomic region across more than one species, be differentially expressed in the same direction across species and have a free energy at least below the 1st Quantile of the free energy distribution of the randomized sequences (Dataset EV10). None of the putative sRNA met these criteria.

## Gut protein extraction, protein gels and enzymatic activity assays

### Protein extraction

To obtain protein extracts, guts were thawed on ice and pooled together (3 guts of adult females with symbiotic organs attached and 4 guts of larvae for *D. versicolorea*, 3 guts from adult females and 3 guts from larvae for *M. mutica*, respectively). The guts were homogenized with three metal beads for 3 min at 50 Hz using the TissueLyser LT (Qiagen, Venlo, The Netherlands). The samples were centrifuged at room temperature for 5 min at $21,300 \times g$. The supernatant was subjected to desalting and protein preparation using the Zeba Desalt Spin Columns with a 7 kDa cutoff (Thermo Fisher Scientific) according to the manufacturer's protocol. Laemmli buffer (BioRad, Hercules, CA, USA), beta-mercaptoethanol was prepared according to the manufacturer's instructions and mixed 1:1 with the protein extract prepared above.

Samples were boiled at 95 °C for 5 min and loaded into a mini-Protean 3 gel (BioRad, Hercules, CA, USA). The samples were allowed to run for 1 h in 10x Tris/Glycine/SDS buffer (BioRad, Hercules, CA, USA) diluted 1:10 in milliQ water at 120 V. The gel was subsequently removed and washed with distilled water for 10 min before being fixed in 40% EtOH and 10% acetic acid for 45 min. Thereafter, the buffer was discarded and the gel washed with distilled water three times for 10 min before being stained with Page Blue Protein Stain (Thermo Fisher, Waltham, MA, USA) overnight. The solution was then discarded, and the gel was washed with distilled water for 2 h. The gel was imaged using the Azure Biosystems (Biozym, Hessisch Oldendorf, Germany). Protein extracts were quantified using the Bradford method (Bradford, 1976) using the Protein assay Dye Concentrate (BioRad, Hercules, CA, USA) according to the manufacturer's protocol with serial dilutions of Albumin (Roth, Karlsruhe, Germany) in an Infinite M200 plate reader (Tecan).

### Thin-layer chromatography

Desalted protein extracts were used for a qualitative assay using TLC as in Kirsch et al, 2014 (Kirsch et al, 2014). Briefly, 51 µg of protein were incubated with 0.4% (w/v) of either Polygalacturonic Acid (from citrus Pectin, Megazyme, Bray, Ireland), Carboxy-methyl Cellulose 4 M (Megazyme, Bray, Ireland) or Xyloglucan (from tamarind seed, Megazyme, Bray, Ireland) in a 20 mM citrate/phosphate buffer pH 5.0 to a total volume of 20 µl at 40 °C for about 16 h. Samples were applied to TLC plates (Silica gel 60, 20 ×20 cm, Merck, Kenilworth, NJ, USA) in 4 2.5 µl steps, and plates were developed ascending with ethyl acetate: glacial acetic acid: formic acid: water (9:3:1:4) for about 120 min. After drying, carbohydrates were stained by spraying plates with 0.2% (w/v) orcinol in methanol: sulfuric acid (9:1), followed by a short heating until spots appeared. The reference standard contained 2 µg each of galacturonic acid (GA) monomer, dimer and trimer and glucose oligomers from mono- to hexamer. Raw datasets can be found in https://doi.org/10.17617/3.1PFBO0.

### 3,5-dinitrosalicylic acid (DNS) assay for enzyme activity quantification

For the DNS assay, 4 µg of protein extracts from five pools of three guts of *Donacia versicolorea* larvae or adults (Dataset EV3), were used to measure pectinase activity against the dialyzed Polygalacturonic Acid (from citrus Pectin, Megazyme, Bray, Ireland) as previously established (Kirsch et al, 2016, 2019) (Dataset EV8).

## Symbiotic organ imaging

Symbiont cell shape was assessed via Fluorescence in situ hybridization (FISH). Samples were processed as in Weiss, 2023 (Weiss, 2023). Briefly, two whole adults of *Donacia thalassina* and *Macroplea mutica* (with elytra, head, and legs removed) as well as two *Macroplea mutica* larvae were fixed in 4% paraformaldehyde (PFA) in 80% tert-butanol. Additionally, dissected symbiotic organs of two *Donacia thalassina* larvae were fixed in 4% PFA. After fixation, samples were dehydrated in a series of ascending butanol concentration (30, 50, 70, 80, 90, and 96%, or 80, 90, and 96% for the samples in 80% Butanol) of tert-Butanol and embedded in Technovit 8100 (Kulzer GmbH, Wehrheim, Germany) according to manufacturer's instructions. From this, transversal histological sections of 8 µm thickness were cut on a Leica

RM 2245 microtome and placed on microscope slides. Before staining, sections were fixed to slides again by applying 4% PFA solution (Roth). The slides were covered with a cover slip and incubated at room temperature (RT) for 15 min after which the slides were submerged in distilled water and the cover slips were removed. The slides were washed by incubating in water for 5 min at RT, twice replacing the water in between. The slides were then dried for 20 min in an incubator at 50 °C. For bacterial staining, 100 µL hybridization mix consisting of hybridization buffer (0.9 M NaCl, 0.02 M Tris/HCl (pH=8), 0.01% SDS), 0.5 µM diluted fluorescently labeled oligonucleotide probes to mark general bacteria (EUB784, Table EV1) or the Donaciinae symbionts specifically (Don_sym_FISH_154, Table EV1) and 5 µg/mL DAPI for DNA staining were applied to each slide which were then covered with a glass cover slip. After hybridizing at 50 °C overnight in a humid box, the cover slips were removed and the slides washed by submerging them in wash buffer (0.1 M NaCl, 0.02 M Tris/HCl (pH=8), 5 mM EDTA, and 0.01% SDS) overnight at 50 °C. An additional washing step was performed in distilled water at 50 °C for 20 min, and finally 30 µL of ProLong Diamond Antifade Mountant (Thermo Fisher Scientific, Waltham, MA, USA) or VECTASHIELD Antifade Mounting Medium (Biozol, Eching, Germany) was applied to each slide and a glass cover slip sealed the sample.

Imaging was done with a Leica THUNDER imager Cell Culture 3D equipped with a K5 monochrome camera (Leica, Wetzlar, Germany). The signals for DAPI, background, Cy3 and Cy5 were obtained with the 390-nm, 475-nm, 555-nm and 635-nm, EFW LED8 respectively, at 50% (Cy3, Cy5, background) or 5% power (DAPI), the DFT51010 filter cube, and 590-nm, 642-nm, 535-nm, and 460-nm fast emission filters, respectively. The Leica Thunder images were processed in the Leica Application Suite X software (Leica), using the large volume computational clearing algorithm with standard settings (data available at https://doi.org/10.17617/3.VQABI2). Images were then further processed using FIJI (Schindelin et al, 2012). Briefly, the brightness and contrast histograms of the Cy5 and DAPI channels were automatically adjusted for each image in the main and supplemental figures, and a scale bar for 20 µm was added.

## Data availability

The raw sequencing datasets produced in this study are available in the European Nucleotide Archive (ENA) under project number PRJEB73718. Annotated symbiont genomes and imaging data are in the Edmond open data repository of the Max-Planck Society under https://doi.org/10.17617/3.VQABI2 and https://doi.org/10.17617/3.1PFBO0.

The source data of this paper are collected in the following database record: biostudies:S-SCDT-10_1038-S44319-025-00525-2.

## Peer review information

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

## Acknowledgements

The authors thank Domenica Schnabelrauch for the Sanger sequencing used for barcoding of *Donacia versicolorea* larvae and Bianca Wurlitzer for support with the enzymatic assays. We are grateful to Bruno Hüttel and the team at the Max-Planck-Genome-Centre Cologne for the RNA-Seq, and Yannick Pauchet for his input on the interpretation of the host differentially expressed CAZymes. The authors thank Benjamin Weiss for the embedding and sectioning of samples for FISH and, together with Tobias Engl, for input on the FISH protocol and imaging. We gratefully acknowledge financial support from the GenEvo Research Training Group funded by the German Research Foundation (DFG) GRK2526/1 (to AC and MK), the Max-Planck Society (to AC, SW, RK, HV, and MK), and the European Research Council (ERC CoG 819585 SYMBeetle to MK).

## Author contributions

**Ana S P Carvalho**: Conceptualization; Data curation; Software; Formal analysis; Investigation; Visualization; Methodology; Writing—original draft; Writing—review and editing. **Sinah T Wingert**: Investigation; Methodology. **Roy Kirsch**: Investigation; Methodology; Writing—review and editing. **Heiko Vogel**: Investigation; Methodology; Writing—review and editing. **Gregor Kölsch**: Resources. **Martin Kaltenpoth**: Conceptualization; Resources; Supervision; Funding acquisition; Project administration; Writing—review and editing.

Source data underlying figure panels in this paper may have individual authorship assigned. Where available, figure panel/source data authorship is listed in the following database record: biostudies:S-SCDT-10_1038-S44319-025-00525-2.

## Funding

## Disclosure and competing interests statement

The authors declare no competing interests.

# Expanded View Figures

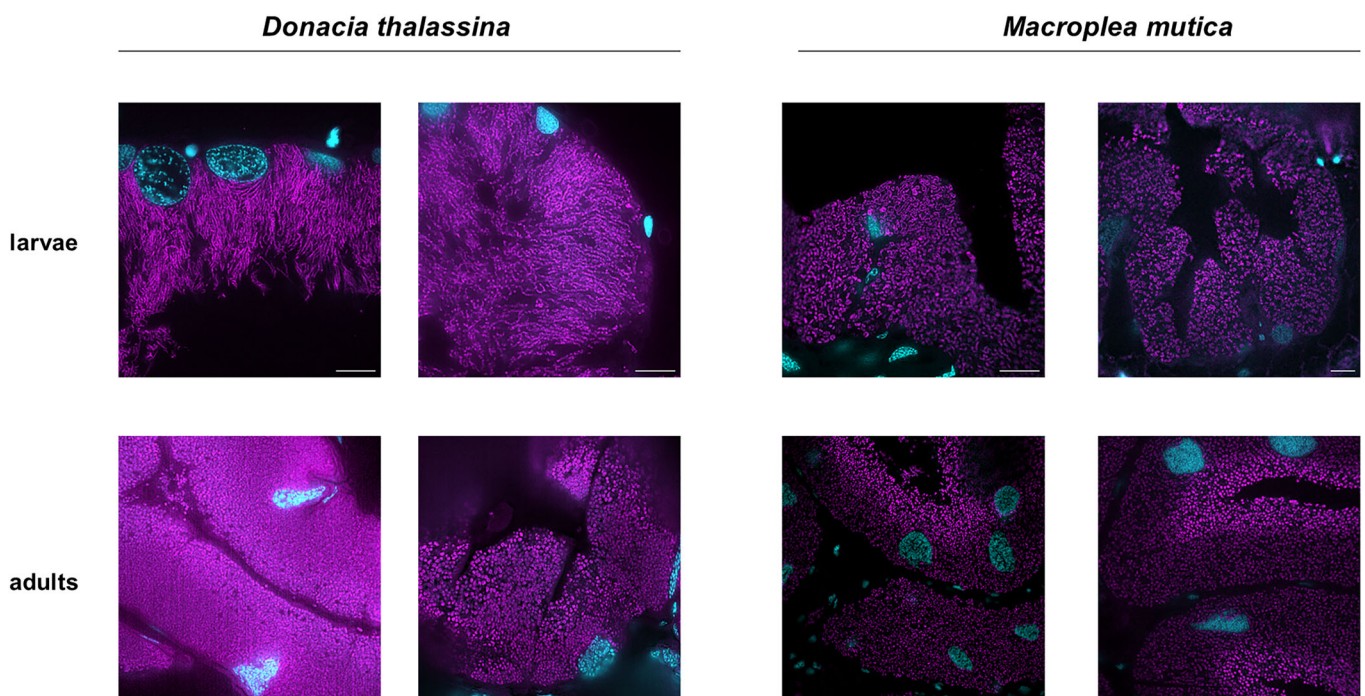

**Figure EV1.  FISH for symbDTHA and symbMMUT in larvae and adults of the respective hosts.**

Symbionts in the symbiotic organs of 2 larvae and 2 adults of *D. thalassina*, and 2 larvae and 2 adults of *M. mutica*. Overlay of a probe specific to Donaciinae symbionts (magenta) and DNA staining with DAPI (cyan). Scale bars correspond to 20 µm. Raw datasets can be found in https://doi.org/10.17617/3.VQABI2.

