## [Peer Review File · EMBO Reports]

Symbionts with eroded genomes adjust gene expression according to host life-stage and environment

Ana Carvalho, Sinah Wingert, Roy Kirsch, Heiko Vogel, Gregor Kölsch, and Martin Kaltenpoth

Corresponding author(s): Martin Kaltenpoth (kaltenpoth@ice.mpg.de)

Review Timeline:

Submission Date:	10th Mar 25
Editorial Decision:	20th Mar 25
Revision Received:	20th Mar 25
Editorial Decision:	22nd Apr 25
Revision Received:	4th Jun 25
Accepted:	25th Jun 25

Transaction Report: A revised version of this manuscript was transferred to EMBO reports following peer review at the EMBO Journal.

Referee #1:

This study on a small genome bacterial symbiont of beetles shows some clearcut changes in gene expression that are linked to two major life stages of the host. Given what is known of the functions of these symbionts during the larval and adult stages, the changes generally make sense, especially the increased expression of amino acid biosynthesis pathways in larvae and the increased production of pectinases in adults.

Most of the paper is based on interpreting transcriptomes in view of the differences between the host life stages. A central question is how these regulatory changes are achieved despite the lack of most regulatory proteins, a feature observed across small genome endosymbionts in general. In the case of *Buchnera* in aphids, which is probably the system in which gene expression has been examined most, published studies don't show much in the way of changes in gene expression, in contrast to this case of the reed beetle symbionts.

The puzzle of how regulatory control is effected is not answered, though some speculations are provided in the Discussion, such as transcription attenuation, which is used in some other bacteria, though they generally also have transcription repressor systems also.

There are some highly relevant papers on the issue of regulation in reduced genomes that are not mentioned:

Hansen & Degan 2014 "Widespread expression of conserved small RNAs in small symbiont genomes" ISME J.

(gives suggestive evidence for a role of post-transcriptional regulation by sRNAs such as antisense RNAs.)

Güell et al. 2009 Transcriptome complexity in a genome-reduced bacterium. Science (shows that *Mycoplasma* uses noncoding, mostly antisense RNAs to regulate other genes, plus many alternative transcripts.

Because this is a challenging system in which genetic tools are lacking for the bacteria (which cannot be cultured), it is not possible to verify these speculations about underlying mechanisms. The findings are nonetheless valuable in providing a convincing example of coordination between the needs of the host and the gene expression of the symbiont. The variation among host species is also interesting.

However, sometimes the conclusions are stated too strongly, implying that causation has been demonstrated when it hasn't. I appreciate that it may be impossible (at least currently) to do the definitive experiments. Still it is important not to overstate support. One example is the final sentence of the Abstract: "Thus, reed beetle symbionts use their few transcription factors to respond to the host's environment, highlighting the regulatory potential of long-term coevolved symbionts despite severely reduced genomes." The basis for the regulation is not yet clear. Small ncRNAs and other mechanisms might be part of the underlying mechanisms.

Even the title is not necessarily true, based on the data presented: perhaps it is some host molecule that is responsible for regulation. More and more instances are being found of nucleic acids, proteins or other molecules moving between symbiotic organisms, enabling one to influence cellular processes in the partner.

Aside from these reservations, I found this to be a clearly presented paper with nice illustrations showing adjustment of symbiont functions to fit host needs.

Referee #2:

In this manuscript "Tiny but resourceful: A severely eroded genome regulates symbiont shape and function" Carvalho et. al. performed transcriptional profiling using RNA-seq to assess whether an insect symbiont with a highly reduced genome (*Candidatus Macrolepicola muticae*), associated with reed beetles (Coleoptera: Chrysomelidae; Donaciinae), can actively regulate its gene expression in response to environmental and host lifestyle changes. Notably, despite this symbiont retaining only 4-5 transcription factors, the authors identified differential regulation of genes potentially associated with temperature responses and varying nutritional demands (e.g., amino acid metabolism and pectinases) between host life stages. The authors also attempt to corroborate their findings with microscopic analysis and enzymatic assays, for example to support the differences in metabolic requirements between larval and adult host stages. Although there is a clear lack of consistency in the expression patterns among the four symbiont strains examined, the authors suggest that the symbiont is capable of actively regulating its gene expression. It is generally believed that transcriptional regulation in obligate mutualistic endosymbionts with highly reduced genomes is minimal. This work contributes to previous studies attempting to shed light on how obligate endosymbionts with highly reduced genomes can respond to environmental or host changes to maintain nutritional and

functional benefits for their hosts.

Overall, the manuscript is well-written. However, some statements in the discussion appear somewhat unnuanced and are not always clearly connected to the findings (see also specific comments below). Additionally, some sections lack methodological details.

Major criticism

1. One of my main criticisms concerns a methodological aspect of this manuscript. The authors aimed to investigate the transcriptional responses of Donaciinae symbionts to temperature differences by raising *D. marginata* larvae under cold and warm conditions for one month before dissecting and extracting RNA from the symbiotic organs. However, according to their methods (lines 533-538), the larvae were placed in ice-cold water for 5 minutes prior to dissection and RNA extraction. Given how rapidly some transcriptional responses occur in bacteria, it is difficult to believe that this treatment did not affect the observed transcriptional responses-particularly for larvae reared under warm conditions. RNA-seq experiments are highly sensitive to sample handling and treatment. Could this procedure have introduced artifacts into the data?

2. The authors suggest that the Donaciinae symbiont is capable of regulating its gene expression under different environmental conditions and host life stages, despite having a limited transcription factor (TF) repertoire. This might be true in principle. However, given the inconsistency in differential expression across the four symbionts examined, I wonder whether the observed patterns of gene expression could be purely stochastic (e.g., resulting from random changes in background expression) rather than indicative of active regulation in response to environmental or life stage changes. Even among phylogenetically closely related symbionts (e.g., *D. marginata* and *D. thalassina*), there appears to be no consistent pattern in gene expression.

3. Moreover, there is evidence from various systems that symbionts with highly reduced genomes can regulate their gene expression using small RNAs (sRNAs) (see, for example, PMID: 29134727 and PMID: 31744912). This is not particularly discussed or mentioned in the manuscript. Are there any indications that Donaciinae symbionts express small RNAs that could potentially regulate gene expression at the post-transcriptional level? Did the experimental design for the RNA sequencing allow the identification of small RNAs?

4. It is not clear to me why the transcriptional response to temperature differences was assessed only in one of the host species (*D. marginata*) and specifically at the larval stage. Although the authors later mention in the Results section that larval development spans multiple seasons and environmental temperatures, I believe a clearer justification for this choice should be provided in the Introduction

5. While the discussion provide some interesting interpretation of the presented results, the connection between some of the sections and the findings is not entirely clear, at least to me. For example, a large part of the section (Regulation of amino acid biosynthesis and coordination with carbohydrate metabolism) discusses the role of ppGpp and transcriptional attenuation in microbial gene regulation. Although these points might be suited to a broader review, their relevance within the context of the current results is not always clear and some of the interpretations appear quite speculative. I think focusing the discussion more closely on the results would enhance the manuscript's coherence. In addition, the order of the discussion sections feels somewhat confusing. For example, the section starting at line 344 (Regulation of amino acid biosynthesis and coordination with carbohydrate metabolism) is related to the previous section at line 288 (Transcriptional regulation of symbiont-provided benefits). However, the two sections are separated by the short paragraph on lines 330-342. Would it be possible to move this paragraph to an earlier position, perhaps before line 249?

6. Accession numbers for the RNA-seq raw reads, transcriptome assemblies and barcoding sequencing data are not provided.

Minor comments

Line 76. Replace "(11)" with a citation.

Line 76. Perhaps use "conform" instead of "go along".

Line 99. Figure 1 is not cited in this paragraph or elsewhere in the text.

Line 153. How similar are the two pectinases in symbMMUT?

Lines 216-217 (Figure 3). Differential expression is defined differently here than in Figures 1 & 2. Please correct.

Line 265. Leave a space after "factor".

Lines 304-305. Lack of homogeneity is also observed in the larval stages.

Line 311. Change "Figure A and B" to "Figure 3 A and B".

Lines 322 - 328. The justification here is not clear to me and the assumption seems a bit vague, especially without supporting citations. Why only FNR? Are there other TFs in GH28

encoding symbionts? Please also define once FNR and CRP .

Line 323. If data are not shown, can the authors indicate which data (figure and/or table) from Reis et al. they are referring to?

Line 338. Use "regulate" instead of "induce"

Line 346. What about the other TFs e.g. the global regulators CRP/FNR are known to be involved in the regulation of several amino acid biosynthetic processes.

Lines 428 - 430. What do you mean when you say that this mechanism acquired by multiple hosts? Please clarify or rephrase. Also "shape-changing symbiont" doesn't read well. How about "form-changing" or "form-shifting" or just omit.

Lines 434 - 438. Is *bolA* expressed differentially between life stages in all 4 strains?

Lines 439 - 460. Again, I feel I'm missing the connection with the current findings.

Line 535. How many pools? How many individuals per pool? Replicates???

Line 541. Which tissue lyser was used? Specify.

Line 545. I think the "DNA Barcoding" section should be combined with the "Insect Collection ..." section to give a clearer understanding of how the samples were collected and identified. Also, have these Sanger sequences already been deposited in GenBank?

Line 551. Why were different primers used for larvae? Please clarify.

Line 656. Why is there such a discrepancy in the number of contigs obtained?

Line 657. Based on what identity cutoff?

Line 661. Replace "genome assembly" with "transcriptome assembly"

Line 664. What do the authors mean by artificial sequences?

Lines 664 - 666. Please provide details of the analysis. What do you mean by "Only enzymes identified with HMMER were included"? What is HMMER? Also consider moving these lines to the next section as they are part of the annotation.

Line 677. Add a space after "R".

Line 681. Add a space after "apeglm"

Figure 1 and Figure 2: I think it would be useful to highlight certain genes that are discussed in the manuscript (e.g. transcription factors, degrading enzymes etc.).

Figure 5 is not cited in the main text.

The Supplementary Figure 1 is not cited in the main text.

Dear Dr. Kaltenpoth

Thank you for the transfer of your research manuscript to our journal. I apologize for my delayed response, which is due to reduced staffing and a high number of submissions in the past week.

My colleague at The EMBO Journal, Cornelius Schneider, had informed you that we are interested to consider your manuscript for publication at EMBO Reports, if the referee concerns are addressed in full.

Just to reiterate the most important points: It will be essential to clarify whether the cold-ice treatment before RNA extraction has had an effect on gene expression and whether this effect was more pronounced in larvae reared at higher temperatures. Conclusions on the causal role of transcription factors in changing gene expression must be toned down and overstatements be avoided. Both referees refer to earlier studies that provided evidence that other symbionts with highly reduced genomes regulate their gene expression using small RNAs. Please address this concern experimentally by either mining your dataset for the expression of small RNAs or by providing additional data on their expression in the symbionts. Moreover, the concern regarding the potentially stochastic nature of gene expression across the four symbionts needs to be addressed and clarified.

You informed me that you have meanwhile fully revised your study and are ready to upload the revised manuscript. Depending on the nature of the revisions, it will be sent back to one or both of the referees for a quick check.

For the sake of time, please upload the study as it stands and we can introduce the required formatting issues later.

- But we do need a Data Availability section that refers to data deposited in public databases. We need the accession number and the URL that resolves directly to the dataset (not just the database).

- We need individual figure files.

- We need a filled Author Checklist. You can download it from our Guide to Authors (<<https://www.embopress.org/page/journal/14693178/authorguide>>). Please insert information in the checklist that is also reflected in the manuscript. The completed author checklist will also be part of the RPF.

- Datasets are called Dataset EV#. They need a legend in a separate tab of the .xls (or .csv) file.

- Please be prepared to generate a Reagents and Tools Table (point 5) and to combine the Results and Discussion section if your manuscript does not exceed 5 main figures. This can also be done at the final revision stage.

- As discussed, we need the source data (minimally modified original data). You can either submit the data as one folder per figure with subfolders for each panel or you deposit it on a publicly available database, such as Biostudies. We need all the quantifications in form of .xls or .csv files, the blots in Figure 3C, D, the confocal images shown in Figure 4 (original format incl. metadata is preferred, but the raw image with the two channels is also acceptable).

Please find below the general formatting guidelines for revised manuscripts.

*******IMPORTANT NOTE:**

We perform an initial quality control of all revised manuscripts before re-review. Your manuscript will FAIL this control and the handling will be delayed IN CASE the following APPLIES:

1) A data availability section providing access to data deposited in public databases is missing. If you have not deposited any data, please add a sentence to the data availability section that explains that.

2) Your manuscript contains statistics and error bars based on $n=2$. Please use scatter blots in these cases. No statistics should be calculated if $n=2$.

When submitting your revised manuscript, please carefully review the instructions that follow below. Failure to include requested items will delay the evaluation of your revision. *****

2) individual production quality figure files as .eps, .tif, .jpg (one file per figure).

Please download our Figure Preparation Guidelines (figure preparation pdf) from our Author Guidelines pages <https://www.embopress.org/page/journal/14693178/authorguide> for more info on how to prepare your figures.

4) a complete author checklist, which you can download from our author guidelines (<<https://www.embopress.org/page/journal/14693178/authorguide>>). Please insert information in the checklist that is also reflected in the manuscript. The completed author checklist will also be part of the RPF.

5) Please note that all corresponding authors are required to supply an ORCID ID for their name upon submission of a revised manuscript (<<https://orcid.org/>>). Please find instructions on how to link your ORCID ID to your account in our manuscript tracking system in our Author guidelines (<<https://www.embopress.org/page/journal/14693178/authorguide#authorshipguidelines>>)

6) We replaced Supplementary Information with Expanded View (EV) Figures and Tables that are collapsible/expandable online. A maximum of 5 EV Figures can be typeset. EV Figures should be cited as "Figure EV1, Figure EV2" etc... in the text and their respective legends should be included in the main text after the legends of regular figures.

7) Please include a dedicated "Data Availability" section at the end of the Methods (suggested wording: "The [structural coordinates | microarray | mass spectrometry] data from this publication have been deposited to the [name of the database] database [URL] and assigned the identifier [accession | permalink | hashtag]."). Should this not apply, this should still be stated as "This study includes no data deposited in external repositories."

Additional information on source data and instruction on how to label the files are available <<https://www.embopress.org/page/journal/14693178/authorguide#sourcedata>>.

10) Figure legends and data quantification:

- the name of the statistical test used to generate error bars and P values,
 - the EXACT p-values,
 - the number (n) of independent experiments (please specify technical or biological replicates) underlying each data point,
 - the nature of the bars and error bars (s.d., s.e.m.)
- If the data are obtained from n {less than or equal to} 5, show the individual data points in addition to the SD or SEM.
- If the data are obtained from n {less than or equal to} 2, use scatter blots showing the individual data points.

11) Our journal encourages inclusion of *data citations in the reference list* to directly cite datasets that were re-used and obtained from public databases. Data citations in the article text are distinct from normal bibliographical citations and should directly link to the database records from which the data can be accessed. In the main text, data citations are formatted as follows: "Data ref: Smith et al, 2001" or "Data ref: NCBI Sequence Read Archive PRJNA342805, 2017". In the Reference list, data citations must be labeled with "[DATASET]". A data reference must provide the database name, accession number/identifiers and a resolvable link to the landing page from which the data can be accessed at the end of the reference. Further instructions are available at <<https://www.embopress.org/page/journal/14693178/authorguide#referencesformat>>.

12) All Materials and Methods need to be described in the main text using our 'Structured Methods' format. According to this format, the Methods section includes a Reagents and Tools Table (listing key reagents, experimental models, software and relevant equipment and including their sources and relevant identifiers) followed by a Methods and Protocols section describing the methods, ideally using a step-by-step protocol format. The aim is to facilitate adoption of the methodologies across labs. Please download and fill our Reagents and Tools Table template (.docx), which you can find in our author guidelines: <https://www.embopress.org/page/journal/14693178/authorguide#structuredmethods>.

13) As part of the EMBO publication's Transparent Editorial Process, EMBO Reports publishes online a Review Process File to accompany accepted manuscripts. This File will be published in conjunction with your paper and will include the referee reports, your point-by-point response and all pertinent correspondence relating to the manuscript.

Kind regards,

Authors' responses are given below the reviewers' comments (in green font, preceded by ">>>")

Editor's comments

I have discussed your study and the referee reports with my colleague Martina Rembold at EMBO Reports who would be interested to consider your manuscript for publication at EMBO Reports, if the referee concerns are addressed in full. It will be essential to clarify whether the cold-ice treatment before RNA extraction has had an effect on gene expression and whether this effect was more pronounced in larvae reared at higher temperatures. Conclusions on the causal role of transcription factors in changing gene expression must be toned down and overstatements be avoided. Both referees refer to earlier studies that provided evidence that other symbionts with highly reduced genomes regulate their gene expression using small RNAs. Please address this concern experimentally by either mining your dataset for the expression of small RNAs or by providing additional data on their expression in the symbionts. Moreover, the concern regarding the potentially stochastic nature of gene expression across the four symbionts needs to be addressed and clarified. If you are interested in this option, please submit the manuscript to EMBO Reports via the transfer link below. Upon transfer, Martina will invite revision of your manuscript asking you to address the points above. Please feel free to contact Martina at m.rembold@emboreports.org if you have any questions about the transfer, EMBO Reports or its policies. Please note that no reformatting of the manuscript is necessary for the transfer.

>>> We thank the editor and the reviewers for their evaluation of our manuscript. We appreciate the time they have taken to provide critical feedback, which helped us to improve our manuscript. Below we address the reviewers' comments in detail, including the points highlighted by the editor as being particularly important. In brief, we have addressed the four main points identified by the editor as follows (details are provided in the responses to the reviewers' comments):

- Ice-cold treatment: this was actually an oversight on our part in the description of the methods, as we failed to indicate that the warm-adapted larvae did not experience the cold treatment before or during dissection. The cold treatment was only done for the cold-adapted larvae, as well as for adults and larvae of the other three species, but not for the warm-adapted *D. marginata* larvae. We apologize for this mistake on our side.
- Causal role of transcription factors: Although plausible based on the extensive literature on the role of transcription factors in the closely related model species *E. coli*, we have indeed not directly demonstrated the role of the transcription factors in gene regulation. Hence, we carefully revised the manuscript to tone down our conclusions. We also included a disclaimer in the discussion to clarify that the role of the TFs needs experimental validation (or support from comparative analyses in other symbioses, given the difficulties in experimentally and genetically manipulating most intracellular symbioses): "Here, we derive hypotheses on the regulatory pathways by comparison of the predicted TFs, the upregulated genes, and the genomes of Donaciinae symbionts with the *E. coli* databases RegulonDB and EcoCyc. It needs to be stressed that our hypotheses on the possible roles of transcription factors in the

Donaciinae symbionts are based on the gene expression profiles and still require experimental validation. However, considering the challenges associated with experimental and genetic manipulation of unculturable symbionts, we hope that our findings and conclusions will encourage comparative approaches across different symbioses that can provide valuable insights into the molecular regulation of symbiont physiology and metabolism.”

- **Small RNAs:** We have analyzed our dataset for sRNAs, which was possible because we used a stranded library for RNAseq, and because many bacterial sRNAs would likely be captured in a library enriched for mRNA. While we identified candidate sRNAs, none of them shared a consistent host life-stage specific differential expression pattern across species. We have added the data to the manuscript and discuss their relevance. Also, we have added references (including the ones suggested by the reviewers) on the role of sRNAs in symbiotic associations.
- **The potentially stochastic nature of gene expression across the four symbionts:** For each species, the gene expression analysis is based on 3-6 replicates per life stage and a robust statistical analysis, making random effects very unlikely (<0.05 per gene, by definition of the adjusted p-value). We have also revised Figure 2 to show more clearly the consistency in the upregulation of amino acid biosynthesis pathways in the larvae of all four species (notably, aromatic amino acids, branched chain amino acids, and histidine). Furthermore, we added KEGG enrichment analyses that consistently revealed amino acid biosynthesis genes as enriched among the upregulated genes in larvae, whereas no functional category was upregulated for any of the adults. Together, these findings in our view provide strong and convincing evidence for the regulation of symbiont gene expression according to the host's life-stage specific demands in nutrients, and the patterns are inconsistent with random changes.

Referee #1:

This study on a small genome bacterial symbiont of beetles shows some clearcut changes in gene expression that are linked to two major life stages of the host. Given what is known of the functions of these symbionts during the larval and adult stages, the changes generally make sense, especially the increased expression of amino acid biosynthesis pathways in larvae and the increased production of pectinases in adults.

Most of the paper is based on interpreting transcriptomes in view of the differences between the host life stages. A central question is how these regulatory changes are achieved despite the lack of most regulatory proteins, a feature observed across small genome endosymbionts in general. In the case of *Buchnera* in aphids, which is probably the system in which gene expression has been examined most, published studies don't show much in the way of changes in gene expression, in contrast to this case of the reed beetle symbionts.

The puzzle of how regulatory control is effected is not answered, though some speculations are provided in the Discussion, such as transcription attenuation, which is used in some other bacteria, though they generally also have transcription repressor systems also.

There are some highly relevant papers on the issue of regulation in reduced genomes that are not

mentioned:

Hansen & Degnan 2014 "Widespread expression of conserved small RNAs in small symbiont genomes" ISME J.

(gives suggestive evidence for a role of post-transcriptional regulation by sRNAs such as antisense RNAs.)

Güell et al. 2009 Transcriptome complexity in a genome-reduced bacterium. Science
(shows that Mycoplasma uses noncoding, mostly antisense RNAs to regulate other genes, plus many alternative transcripts.

Because this is a challenging system in which genetic tools are lacking for the bacteria (which cannot be cultured), it is not possible to verify these speculations about underlying mechanisms. The findings are nonetheless valuable in providing a convincing example of coordination between the needs of the host and the gene expression of the symbiont. The variation among host species is also interesting.

However, sometimes the conclusions are stated too strongly, implying that causation has been demonstrated when it hasn't. I appreciate that it may be impossible (at least currently) to do the definitive experiments. Still it is important not to overstate support. One example is the final sentence of the Abstract: "Thus, reed beetle symbionts use their few transcription factors to respond to the host's environment, highlighting the regulatory potential of long-term coevolved symbionts despite severely reduced genomes." The basis for the regulation is not yet clear. Small ncRNAs and other mechanisms might be part of the underlying mechanisms.

Even the title is not necessarily true, based on the data presented: perhaps it is some host molecule that is responsible for regulation. More and more instances are being found of nucleic acids, proteins or other molecules moving between symbiotic organisms, enabling one to influence cellular processes in the partner.

Aside from these reservations, I found this to be a clearly presented paper with nice illustrations showing adjustment of symbiont functions to fit host needs.

>>> **Many thanks for the overall positive evaluation of our manuscript and the constructive suggestions to further improve its quality. We particularly appreciate the reviewer's acknowledgement of the challenges associated with working on a non-model organism with unculturable symbionts. Based on the reviewer's comments, we have toned down the conclusions from our results in several places where we thought this is necessary (including in the title, abstract, discussion and conclusion), and we have added references to the suggested citations. We also addressed the comment on the potential involvement of small RNAs in post-transcriptional regulation by analyzing our dataset for possible regulatory sRNAs and by discussing the evidence provided in previous studies from other groups on sRNAs in host-symbiont interactions (please refer to our more detailed response on this point in the responses to reviewer #2's comments).**

Referee #2:

In this manuscript "Tiny but resourceful: A severely eroded genome regulates symbiont shape and function" Carvalho et. al. performed transcriptional profiling using RNA-seq to assess whether an insect

symbiont with a highly reduced genome (*Candidatus Macropleicola muticae*), associated with reed beetles (Coleoptera: Chrysomelidae; Donaciinae), can actively regulate its gene expression in response to environmental and host lifestyle changes. Notably, despite this symbiont retaining only 4-5 transcription factors, the authors identified differential regulation of genes potentially associated with temperature responses and varying nutritional demands (e.g., amino acid metabolism and pectinases) between host life stages. The authors also attempt to corroborate their findings with microscopic analysis and enzymatic assays, for example to support the differences in metabolic requirements between larval and adult host stages. Although there is a clear lack of consistency in the expression patterns among the four symbiont strains examined, the authors suggest that the symbiont is capable of actively regulating its gene expression. It is generally believed that transcriptional regulation in obligate mutualistic endosymbionts with highly reduced genomes is minimal. This work contributes to previous studies attempting to shed light on how obligate endosymbionts with highly reduced genomes can respond to environmental or host changes to maintain nutritional and functional benefits for their hosts. Overall, the manuscript is well-written. However, some statements in the discussion appear somewhat unnuanced and are not always clearly connected to the findings (see also specific comments below). Additionally, some sections lack methodological details.

>>> Thank you for the feedback and the constructive criticism. In our revision of the manuscript, we toned down statements throughout the abstract, discussion and conclusion. Regarding the lack of consistency in gene expression patterns across species, we provide a detailed response below, and we address the other points in detail as well.

Major criticism

1. One of my main criticisms concerns a methodological aspect of this manuscript. The authors aimed to investigate the transcriptional responses of Donaciinae symbionts to temperature differences by raising *D. marginata* larvae under cold and warm conditions for one month before dissecting and extracting RNA from the symbiotic organs. However, according to their methods (lines 533-538), the larvae were placed in ice-cold water for 5 minutes prior to dissection and RNA extraction. Given how rapidly some transcriptional responses occur in bacteria, it is difficult to believe that this treatment did not affect the observed transcriptional responses-particularly for larvae reared under warm conditions. RNA-seq experiments are highly sensitive to sample handling and treatment. Could this procedure have introduced artifacts into the data?

>>> Thank you for pointing this out, and our apologies for the oversight on our part, there was in fact an error in the description of our methods. All samples were cooled down (adult beetles immobilized at -20°C for 5 min, larvae placed in ice cold water for 5 min) before dissection, except for the warm-adapted larvae, which were kept at room temperature prior to and during dissection, in order to avoid a cold-shock in the warm-adapted individuals. We missed out on stating this in the methods section of the original manuscript, and we apologize again for this mistake. So these individuals never experienced cold conditions, whereas the cold-adapted larvae were maintained in the cold and then also kept cold during dissection and ice-cold water is also something they would experience in the

winter in nature. Therefore, we think that the comparison between cold- vs. warm-adapted larvae is valid. However, one – in our view minor – experimental flaw is that the *D. marginata* adults experienced a short cold shock (we need to immobilize the adult beetles by a short exposure to -20°C for dissection), whereas the larvae they are compared to were not (these are the warm-adapted larvae from the temperature experiment). This was a specific issue only for *D. marginata*, while for the three other species, both adults and larvae were dissected under cold conditions. However, as we see very similar results for the life-stage specific symbiont gene expression in *D. marginata* as for the other three species, we are confident that the 5 min cold exposure of the adults did not have a strong impact on symbiont gene expression.

We corrected the methods section and clarified these points in the revised manuscript.

2. The authors suggest that the Donaciinae symbiont is capable of regulating its gene expression under different environmental conditions and host life stages, despite having a limited transcription factor (TF) repertoire. This might be true in principle. However, given the inconsistency in differential expression across the four symbionts examined, I wonder whether the observed patterns of gene expression could be purely stochastic (e.g., resulting from random changes in background expression) rather than indicative of active regulation in response to environmental or life stage changes. Even among phylogenetically closely related symbionts (e.g., *D. marginata* and *D. thalassina*), there appears to be no consistent pattern in gene expression.

>>> Symbiont gene expression indeed differs between species, but we disagree with the statement that there appears to be no consistent pattern in gene expression, and for multiple reasons we think it is highly unlikely that the observed patterns are due to random changes in background expression. First, the results presented in Figure 2 are based on a statistical analysis of 3-6 replicate RNAseq samples per species and life stage, and only significantly differentially expressed genes are presented. These are generally accepted procedures for differential gene expression analyses, so it is extremely unlikely that the differential expression between life stages within each species are purely stochastic. Second, considering the phylogenetic distance of the four host species (~67 million years between *Macropoda* and *Donacia*, and ~19 million years between the “closely” related species *D. marginata* and *D. thalassina*, see Kölsch et al. in *Molecular Phylogenetics and Evolution* 48 (2008) 936–952) and the different host plants they have adapted to, it is not entirely surprising to see some differences in gene expression. Interestingly, however, amino acid biosynthetic pathways are consistently upregulated in the larvae of all four species. Some of these pathways are upregulated across species, some only in individual species, but it is striking to see a high degree of overlap in genes involved in the biosynthesis of aromatic amino acids, branched-chain amino acids, and histidine being upregulated in larvae across species. In order to better visualize these patterns, we decided to display all four species separately in Figure 2, which highlights upregulation of amino acid biosynthesis pathway genes in larvae, but not adults, of all four species. Third, to address the comment even more directly, we performed a KEGG enrichment analysis, which confirmed that – while there is no significant enrichment of any KEGG category during host adult stages – the upregulated symbiont

genes during larval stages of all species were enriched in amino acid biosynthesis pathways. We added this analysis as a supplementary dataset and now reference it throughout the manuscript.

3. Moreover, there is evidence from various systems that symbionts with highly reduced genomes can regulate their gene expression using small RNAs (sRNAs) (see, for example, PMID: 29134727 and PMID: 31744912). This is not particularly discussed or mentioned in the manuscript. Are there any indications that Donaciinae symbionts express small RNAs that could potentially regulate gene expression at the post-transcriptional level? Did the experimental design for the RNA sequencing allow the identification of small RNAs?

>>> We thank the reviewer for pointing this out, it is indeed an important point. In both references provided, the effect of differentially expressed sRNAs is proposed to be especially relevant for regulation at the post-transcriptional level. Our research focuses on the potential regulation of transcription specifically as we do not have any proteomics data. However, bacterial sRNAs can affect mRNA stability and degradation, which could potentially be reflected in transcript abundance. We did not design our RNA sequencing specifically to focus on sRNAs, but rather on the mRNAs. However, due to the range of sizes of bacterial sRNAs between 50-500 nucleotides (Storz et al 2004 Controlling mRNA stability and translation with small, noncoding RNAs. *Curr Opin Microbiol.* 7(2):140-4; Boisset et al. 2007 *Staphylococcus aureus* RNAlII coordinately represses the synthesis of virulence factors and the transcription regulator Rot by an antisense mechanism. *Genes Dev.* 21(11):1353-66.), we were able to perform an investigation of potential sRNAs expressed by these symbionts, with the caveat that transcripts below 150 bp will likely be absent.

For this sRNA analysis, we used the APERO software according to Leonard et al. 2019 (maximum width and distance = 10, enrichment = 0.05, minimum read number = 10). We analyzed the merged bam file for each life stage per species. Potential sRNAs below 500 nucleotides and classified as antisense or intergenic were kept in the output of adults and larvae for each species. The results for the different life-stages for each species was merged using GFFcompare and boundaries for potential sRNA regions were determined based on the loci annotation (one locus = one potential sRNA region). The resulting GTF was then used together with the individual bam file for each replicate to count reads mapping to each locus using FeatureCounts. We then performed differential gene expression analysis with DESeq2 between the life-stages of each species. The predicted free energy of differentially expressed sRNAs was compared to that of 100 randomized sequences with the same GC content in length. Additionally, we searched for overlapping differentially expressed sRNAs using a Mauve whole genome alignment. A candidate sRNA should have a predicted free energy lower than at least the 1st quantile of the distribution of the free energy of randomized sequences, be present in at least two of the four species and be differentially expressed in the same direction across species. We defined these criteria in analogy to what was previously done for the identification of sRNAs in *Buchnera* (Thairu et al, (2018). A sRNA in a reduced mutualistic symbiont genome regulates its own gene expression, *Molecular Ecology*). Interestingly, none of the putative sRNAs in the Donaciinae symbionts met these criteria. However, we did find a couple of sRNA in the vicinity of genes that are regulated by sRNAs in *Buchnera*. Namely, we identified a sRNA antisense to *carB* in symbMMUT, a sRNA

antisense to the 3' of *prnB* and 5' of *aroC* in *symbDTHA* and a putative sRNA antisense to 5' *aroC* in *symbMMUT*.

Thus, while our sRNA analysis revealed potential candidates, we did not detect consistently differentially expressed sRNAs across species, and their roles in post-transcriptional regulation remain speculative. Nevertheless, we included the results of these analyses in the manuscript, as they provide interesting new avenues for future research on this and other systems.

In the manuscript, we included the following paragraphs in the results and discussion section, respectively:

“Another potential mechanism of regulation in symbionts with reduced genomes are small non-coding RNAs (sRNA). We searched the transcriptomic data for each life stage of each species for putative sRNAs using APERO (Leonard et al., 2019), by investigating transcripts below 500 nucleotides in length that were classified as either antisense to a coding sequence or located in an intergenic region. By combining the results for both life stages for each species, we obtained 154 to 364 putative sRNA regions. We then analyzed the life stage-specific expression patterns of all putative sRNA, revealing 6, 76, 9, and 11 differentially expressed sRNAs for *D. marginata*, *D. thalassina*, *D. versicolorea*, and *M. mutica*, respectively, which were then analyzed in terms of predicted free energy and genomic localization. To identify a sRNA candidate, we expected a predicted minimum free energy secondary structure below the first quantile of the free energy distribution of 100 randomized sequences of the same length and GC content. Additionally, the same candidate sRNA should be predicted and differentially expressed in the same direction in more than one species. However, none of the putative sRNAs met these criteria (Datasets 15 and 16). Thus, while sRNAs possibly play a role in the regulation of symbiont processes, we could not identify any putative regulatory sRNAs that were shared among several of the symbionts.”

“In other symbionts with a reduced genome such as the aphid endosymbiont *Buchnera*, regulation of symbiont gene expression across host life-stages can also be achieved by small non-coding RNA (sRNA) (Hansen & Degnan, 2014; Thairu et al., 2018; Thairu & Hansen, 2019). sRNAs have been implicated in the regulation of several amino acid biosynthesis pathways. In fact, an antisense sRNA was validated in affecting the translation of *carAB* (Thairu et al., 2018) and several others likely target the branched-chain amino acid biosynthesis, aromatic amino acid biosynthesis, the pentose phosphate pathway, and riboflavin biosynthesis (Thairu & Hansen, 2019). In our analysis of the Donaciinae symbionts, we did not find sRNA candidates that showed consistent life stage-specific expression changes across species. However, some of the putative sRNAs identified with APERO were antisense to the same genes or pathways as the ones found in *Buchnera*. Namely, a sRNA antisense to *carB* in *symbMMUT*, but this gene was not differentially expressed (Supplementary data X). The same is true for an sRNA antisense to the 3'-region of *prnB* and 5'-region of *aroC* in *symbDTHA*, and a putative sRNA antisense to 5'-region *aroC* in *symbMMUT* (Supplementary data X). Additionally, putative sRNAs were identified antisense to the 5'-region of the transketolase gene (*tkt*) across all symbionts. Although there is potential for sRNAs to have a similar regulatory role as in *Buchnera*, further studies pairing transcriptomics focused on small transcripts and proteomics will be necessary

to better understand the potential role of sRNA-mediated regulation of gene expression and both transcriptional and post-transcriptional level in the symbionts of the Donaciinae.”

4. It is not clear to me why the transcriptional response to temperature differences was assessed only in one of the host species (*D. marginata*) and specifically at the larval stage. Although the authors later mention in the Results section that larval development spans multiple seasons and environmental temperatures, I believe a clearer justification for this choice should be provided in the Introduction

>>> The temperature experiment was done with the only species for which we had an established rearing system, as the beetles are very host-plant specific and not easy to rear in the lab. And indeed, the long larval development was the main motivation to do the experiment with larvae. The adults only occur during a few weeks in summer and experience much less of a seasonal variation in temperatures, whereas the larvae can be found year-round in the water. Thus, we set out to test whether symbiont gene expression would differ between larvae kept under “summer” and “winter” temperatures, respectively. Additionally, the larvae are the life stage that has to accumulate biomass, so we expected a stronger impact of temperature on gene expression.

We added the information on why this experiment was done only with *D. marginata* to the appropriate results section and the information on the lengthy life-cycle to the introduction.

5. While the discussion provides some interesting interpretation of the presented results, the connection between some of the sections and the findings is not entirely clear, at least to me. For example, a large part of the section (Regulation of amino acid biosynthesis and coordination with carbohydrate metabolism) discusses the role of ppGpp and transcriptional attenuation in microbial gene regulation. Although these points might be suited to a broader review, their relevance within the context of the current results is not always clear and some of the interpretations appear quite speculative. I think focusing the discussion more closely on the results would enhance the manuscript's coherence. In addition, the order of the discussion sections feels somewhat confusing. For example, the section starting at line 344 (Regulation of amino acid biosynthesis and coordination with carbohydrate metabolism) is related to the previous section at line 288 (Transcriptional regulation of symbiont-provided benefits). However, the two sections are separated by the short paragraph on lines 330-342. Would it be possible to move this paragraph to an earlier position, perhaps before line 249?

>>> Thank you for the feedback. Following the reviewer’s recommendations, we restructured the discussion in order to make it more coherent and closer to the results at hand.

6. Accession numbers for the RNA-seq raw reads, transcriptome assemblies and barcoding sequencing data are not provided.

>>> We indicated in the manuscript that the raw sequencing datasets produced in this study have been uploaded to the European Nucleotide Archive (ENA) under project number PRJEB73718 and will be made publicly available upon acceptance of the manuscript. Of course, we are happy to provide

access to the reviewers if necessary. The barcoding data and transcriptome assemblies will also be made available upon acceptance.

Minor comments

Line 76. Replace "(11)" with a citation.

>>> Done.

Line 76. Perhaps use "conform" instead of "go along".

>>> Done.

Line 99. Figure 1 is not cited in this paragraph or elsewhere in the text.

>>> Done.

Line 153. How similar are the two pectinases in symbMMUT?

>>> We have previously reported that the two pectinases likely have different evolutionary origins, with the plasmid-encoded pectinase clustering with the pectinase of the tortoise leaf beetle symbiont *Stammera capleta* within a clade of proteobacterial and acidobacterial pectinases, and the chromosome-encoded pectinase clustering within a clade of bacterial pectinases with mixed taxonomic affiliations (Supplementary figures 1 and 2 of Reis *et al* 2020). We previously achieved heterologous expression and enzymatic assays of the plasmid-encoded pectinase from *M. mutica* and showed its activity to break down polygalacturonic acid into galacturonan oligomers (Reis *et al* 2020). While heterologous expression of the chromosome-encoded pectinase failed, comparative enzymatic assays with gut extracts of species encoding both vs. only the plasmid-encoded pectinase provided strong evidence that the chromosome-encoded pectinase aids in the break-down of the galacturonan oligomers into monomers. Thus, the two pectinases likely complement each other in the degradation of pectin. It remains unknown why some species lost one copy, whereas others retained both, but we previously speculated that differences in life style or host plant chemistry may have been the driving factors (Reis *et al.* 2020).

Lines 216-217 (Figure 3). Differential expression is defined differently here than in Figures 1 & 2. Please correct.

>>> Done.

Line 265. Leave a space after "factor".

>>> Done.

Lines 304-305. Lack of homogeneity is also observed in the larval stages.

>>> This section has been restructured. Additionally, statements regarding homogeneity of the gene expression were changed throughout.

Line 311. Change "Figure A and B" to "Figure 3 A and B".

>>> Done.

Lines 322 - 328. The justification here is not clear to me and the assumption seems a bit vague, especially without supporting citations. Why only FNR? Are there other TFs in GH28 encoding symbionts? Please also define once FNR and CRP.

>>> Pectinase-encoding symbionts share the same set of transcription factors as non-pectinase-encoding symbionts with one exception: FNR. This putative FNR is the only TF (in fact, the only gene other than the pectinases themselves!) whose presence perfectly correlates with the presence/absence of the pectinases across the symbionts of the 26 reed species we investigated previously (Reis et al 2020. Supplementary data1, FNR in cluster 408, Polygalacturonases in cluster 418). This, in our view, provides strong evidence for a connection between FNR and the pectinases.

Line 323. If data are not shown, can the authors indicate which data (figure and/or table) from Reis et al. they are referring to?

>>> The information was added.

Line 338. Use "regulate" instead of "induce"

>>> Done.

Line 346. What about the other TFs e.g. the global regulators CRP/FNR are known to be involved in the regulation of several amino acid biosynthetic processes.

>>> CRP is capable of affecting the transcription of genes involved in a variety of functions. However, it is best known for the regulation of carbohydrate metabolism. Based on our data, CRP may indeed have an indirect effect on the regulation of amino acid biosynthesis. In our dataset, the PTS genes *ptsH* and *ptsI* are upregulated during the larval stages of *D. marginata* and *D. versicolorea*. Focusing on *smbDVER*, some differentially expressed genes involved in carbohydrate metabolism during larval stages are putatively regulated by CRP, namely *ptsH*, *fbxA*, *pgk* and *tktB*. Additionally, we do observe upregulation of CRP in *symbDVER* during host adult stages. We can thus hypothesize that CRP mediated regulation can lead to upregulation of carbohydrate metabolism, which, as described towards the end of the section "Regulation of amino acid biosynthesis and coordination with carbohydrate metabolism", can lead to an indirect regulation of amino acid biosynthesis. However, many genes have more than one promoter for CRP binding and can be either positively or negatively regulated by CRP depending on external conditions, such as the levels of available cAMP. As for FNR, due to its perfect correlation with the presence/absence of pectinase we think that its role is likely tied to this function rather than to amino acid metabolism.

Lines 428 - 430. What do you mean when you say that this mechanism acquired by multiple hosts? Please clarify or rephrase. Also "shape-changing symbiont" doesn't read well. How about "form-changing" or "form-shifting" or just omit.

>>> This was rephrased for clarification. It now reads as follows: "In another example, *Buchnera*, the machinery for biosynthesis and regulation of the bacterial cell wall consists of a patchwork of symbiont- and host-encoded genes, with the latter deriving from horizontal gene transfer events from different bacteria (Smith et al., 2022)."

Lines 434 - 438. Is bolA expressed differentially between life stages in all 4 strains?

>>> BolA is not differentially expressed by any of the four symbionts. However, since BolA is regulated by phosphorylation (<https://pubmed.ncbi.nlm.nih.gov/32535996/>), identification of BolA regulation can require phosphoproteomics.

Lines 439 - 460. Again, I feel I'm missing the connection with the current findings.

>>> Changes in cell shape appear to be an important feature of insect symbionts, as they are observed across many different systems. However, the regulatory basis of these changes remains largely unknown. This is why we spend this additional paragraph on discussing the implications of cell shape changes across symbioses, in order to make the implications of our findings on possible regulators of cell shape in genome-eroded symbionts accessible to a broader readership.

Line 535. How many pools? How many individuals per pool? Replicates???

>>> As it is stated in the manuscript, the symbiotic organs of each individual were pooled together (larvae contain four organs each, adults two), leading to one pool per individual. We added a table to the methods section to specify the number of individual replicates used in the transcriptomics experiments. The information is additionally accessible from Dataset 11.

Line 541. Which tissue lyser was used? Specify.

>>> Done.

Line 545. I think the "DNA Barcoding" section should be combined with the "Insect Collection ..." section to give a clearer understanding of how the samples were collected and identified. Also, have these Sanger sequences already been deposited in GenBank?

>>> The DNA Barcoding section was moved according to the reviewer's suggestion. The barcoded sequences will be deposited in GenBank and made accessible upon acceptance of the manuscript.

Line 551. Why were different primers used for larvae? Please clarify.

>>> Since amplification efficiency was mixed, especially for *D. versicolorea* samples, a second primer pair was designed to amplify the Donaciinae COI.

Line 656. Why is there such a discrepancy in the number of contigs obtained?

>>> We do not know exactly why there is such a discrepancy. We carefully checked for contamination (e.g. eukaryotic parasites), but found no evidence for it. Rather, it seems that the allelic diversity in *D. versicolorea* is higher than in *M. mutica*, which may have resulted in this discrepancy. Additionally, *D. versicolorea* samples were collected across multiple years. Also, *M. mutica* is a much rarer species, with small and isolated populations, so reduced genetic diversity may be expected.

Line 657. Based on what identity cutoff?

>>> The identity cutoff was 0.9, as it is the default. This was now explicitly stated in the manuscript.

Line 661. Replace "genome assembly" with "transcriptome assembly"

>>> Done.

Line 664. What do the authors mean by artificial sequences?

>>> **We were referring to sequencing reads occurring due to contamination with other organisms/organelles. We clarified this to read: "Transcripts assigned to chloroplasts, bacteria, archaea or fungi were removed from the analysis."**

Lines 664 - 666. Please provide details of the analysis. What do you mean by "Only enzymes identified with HMMER were included"? What is HMMER? Also consider moving these lines to the next section as they are part of the annotation.

>>> **We changed the section's title to include the functional annotation. HMMER is a software for the identification of proteins and/or domains using hidden Markov models, the reference to HMMER website was added as per suggestion of the HMMER user manual.**

Line 677. Add a space after "R".

>>> **Done.**

Line 681. Add a space after "apeglm"

>>> **Done.**

Figure 1 and Figure 2: I think it would be useful to highlight certain genes that are discussed in the manuscript (e.g. transcription factors, degrading enzymes etc.).

>>> **We highlighted the genes that are discussed in the text in bold font. Also, we completely restructured Figure 2 and split it up into the four different species. This helps to identify the commonalities in amino acid biosynthesis upregulation in the larvae.**

Figure 5 is not cited in the main text.

>>> **Thanks for pointing this out. We added several citations to Figure 5 in the discussion and conclusions.**

The Supplementary Figure 1 is not cited in the main text.

>>> **Done.**

Manuscript number: EMBOR-2025-61498V2

Title: Tiny but resourceful: regulation of shape and function in a symbiont with severely eroded genome

Author(s): Ana Carvalho, Sinah Wingert, Roy Kirsch, Heiko Vogel, Gregor Kölsch, and Martin Kaltenpoth

Dear Martin,

Thank you for submitting your revised manuscript to EMBO Reports and for your patience while it was assessed by former referee #2. As you will see from the report below, the referee considers the revised version much improved and recommends publication.

I am therefore writing with an 'accept in principle' decision, which means that I will be happy to accept your manuscript for publication once a few minor issues/corrections have been addressed, as follows.

- Your manuscript will be published in our Reports section, which would require the combination of the Results and Discussion section and a shorter, more focussed discussion.
 - The Conflict of Interest section needs to be renamed to Disclosure and Competing Interests Statement.
 - References: et al needs to be used after 10 author names; DOIs should only be used for preprints and datasets that have not been published yet.
 - Author checklist: please complete column D by choosing the appropriate answer from the pull down menus.
 - Callouts to the following figure panels and datasets are missing and should be added in the text where appropriate: Figure 2A and Figure 2B; Dataset EV2, EV4, EV6-EV10, EV16.
 - Supplementary figure 2 is called out but missing.
 - Supplementary Table 1 is a dataset and needs to be updated to Dataset EV# in all places.
 - Dataset EV3 and EV6 .csv files have only one tab and apparently no data.
 - The legends of the datasets and Suppl. Table 1 need to be removed from the manuscript.
 - Please remove Supplement Figure 1 from the manuscript and provide it as Figure EV1. The legend should be provided in the manuscript file after the main figure legends.
 - Please download and fill our Reagents and Tools Table template (.docx), which you can find in our author guidelines: <https://www.embopress.org/page/journal/14693178/authorguide#structuredmethods>. When submitting your revised manuscript, please do not include the Reagents and Tools Table in the Methods section of the manuscript but upload it as a separate file choosing the file type "Reagent Table". An example of a Method paper with Structured Methods can be found here: <https://www.embopress.org/doi/10.15252/msb.20178071>.
 - The manuscript sections should be in the following order: Title page - Abstract & Keywords - Introduction - Results & Discussion - Methods - Data Availability - Acknowledgments - Disclosure Statement & Competing Interests - References - Figure Legends - (Main Tables with legends if applicable) - Expanded View Figure Legends.
 - Material and Methods should be Methods.
 - Our production/data editors have asked you to clarify several points in the figure legends (see below). Please incorporate these changes in the manuscript and return the revised file with tracked changes with your final manuscript submission.
- A) Statistical test information. Only p-values that are actually shown in the figure panel(s) should (and must) be defined in the legends, all others should be removed from (or added to) the legend. Moreover, we ask for the specification of exact p-values:
- Please indicate what */ **/ ***/ **** represents; if this represents p value(s), please specify the exact p value in the legend(s) 3E, F
 - Please indicate the statistical test used for data analysis in the legends of figures 3A, B, F.
- B) Replicates and error bars:
- Please note that the box plots need to be defined in terms of minima, maxima, centre, bounds of box and whiskers, and percentile in the legend of figure 3F
 - Please note that information related to n is missing in the legend of figure 3F.

C) Data presentation:

- Please note that the asterisk is not defined in the legend of figures 3C, D. This needs to be rectified.

- As a standard procedure we edit the title and abstract to make it more accessible to our general readership. Please note that our titles may not contain punctuation marks.

What about this title: Symbionts with eroded genomes change gene expression according to host life-stage and environment?

- Finally, EMBO Reports papers are accompanied online by

A) a short (1-2 sentences) summary of the findings and their significance,

B) 2-3 bullet points highlighting key results and

C) a schematic summary figure that provides a sketch of the major findings (not a data image).

Please provide the summary figure as a separate file in PNG or JPG format at a size of 550x300-600 pixels (width x height).

Please note that the size is rather small and that text needs to be readable at the final size. Please send us this information along with the revised manuscript.

Once you have made these minor revisions, please use the following link to submit your corrected manuscript:

Link Not Available

If all remaining corrections have been attended to, you will then receive an official decision letter from the journal accepting your manuscript for publication in the next available issue of EMBO reports. This letter will also include details of the further steps you need to take for the prompt inclusion of your manuscript in our next available issue.

Thank you for your contribution to EMBO reports.

Kind regards,

Martina

Martina Rembold, PhD

Senior Editor

EMBO reports

=====

Referee #2:

First of all, I would like to thank the authors for thoroughly and carefully addressing all the reviewers comments from their original submission to the EMBO Journal. I especially appreciate their effort and time in performing the additional analysis on small RNAs and restructuring the figures and text.

I think the revised manuscript is much improved and reads really well. The discussion is also much more concise and clear. This manuscript will be an important addition to the body of knowledge in the field of symbiosis.

I have no further comments.

All editorial and formatting issues were resolved by the authors.

Dr. Martin Kaltenpoth
Max Planck Institute for Chemical Ecology
Insect Symbiosis
Hans-Knöll-Str. 8
Jena, Thuringia 07745
Germany

Dear Martin,

I am very pleased to accept your manuscript for publication in the next available issue of EMBO reports. Thank you for your contribution to our journal.

Kind regards,

Martina
